# Adaptive dating and fast proposals: Revisiting the phylogenetic relaxed clock model

**Jordan Douglas** [1,2]*, **Rong Zhang** [1,2], **Remco Bouckaert** [1,2,3]

**1** Centre for Computational Evolution, University of Auckland, Auckland, New Zealand, **2** School of Computer Science, University of Auckland, Auckland, New Zealand, **3** Max Planck Institute for the Science of Human History, Jena, Germany

* jordan.douglas@auckland.ac.nz

**Data Availability Statement:** The work presented in this article was implemented as a user friendly open source BEAST 2 package with GUI support via BEAUti making it easy to set up an analysis. Instructions for using and installing this package

## Abstract

Relaxed clock models enable estimation of molecular substitution rates across lineages and are widely used in phylogenetics for dating evolutionary divergence times. Under the (uncorrelated) relaxed clock model, tree branches are associated with molecular substitution rates which are independently and identically distributed. In this article we delved into the internal complexities of the relaxed clock model in order to develop efficient MCMC operators for Bayesian phylogenetic inference. We compared three substitution rate parameterisations, introduced an adaptive operator which learns the weights of other operators during MCMC, and we explored how relaxed clock model estimation can benefit from two cutting-edge proposal kernels: the AVMVN and Bactrian kernels. This work has produced an operator scheme that is up to 65 times more efficient at exploring continuous relaxed clock parameters compared with previous setups, depending on the dataset. Finally, we explored variants of the standard narrow exchange operator which are specifically designed for the relaxed clock model. In the most extreme case, this new operator traversed tree space 40% more efficiently than narrow exchange. The methodologies introduced are adaptive and highly effective on short as well as long alignments. The results are available via the open source optimised relaxed clock (ORC) package for BEAST 2 under a GNU licence (https://github. com/jordandouglas/ORC).

## Author summary

Biological sequences, such as DNA, accumulate mutations over generations. By comparing such sequences in a phylogenetic framework, the evolutionary tree of lifeforms can be inferred and historic divergence dates can be estimated. With the overwhelming availability of biological sequence data, and the increasing affordability of collecting new data, the development of fast and efficient phylogenetic algorithms is more important than ever. In this article we focus on the relaxed clock model, which is very popular in phylogenetics. We explored how a range of optimisations can improve the statistical inference of the relaxed clock. This work has produced a phylogenetic setup which can infer parameters related to the relaxed clock up to 65 times faster than previous setups, depending on the

can be found at https://github.com/jordandouglas/ORC.

**Funding:** J.D. and R.B. were funded by Marsden grant 18-UOA-096. The funders had no role in study design, data collection and analysis, decision to publish, or preparation of the manuscript.

**Competing interests:** The authors have declared that no competing interests exist.

dataset. The methods introduced adapt to the dataset during computation and are highly efficient when processing long biological sequences.

## Introduction

The molecular clock hypothesis states that the evolutionary rates of biological sequences are approximately constant through time [1]. This assumption forms the basis of clock-model-based phylogenetics, under which the evolutionary trees and divergence dates of life forms are inferred from biological sequences, such as nucleic and amino acids [2, 3]. In Bayesian phylogenetics, these trees and their associated parameters are estimated as probability distributions [4–6]. Statistical inference can be performed by the Markov chain Monte Carlo (MCMC) algorithm [7, 8] using platforms such as BEAST [9], BEAST 2 [10], MrBayes [11], and RevBayes [12].

The simplest phylogenetic clock model—the strict clock—makes the mathematically convenient assumption that the evolutionary rate is constant across all lineages [4, 5, 13]. However, molecular substitution rates are known to vary over time, over population sizes, over evolutionary pressures, and over nucleic acid replicative machineries [14–16]. Moreover, any given dataset could be clock-like (where substitution rates have a small variance across lineages) or non clock-like (a large variance). In the latter case, a strict clock is probably not suitable.

This led to the development of relaxed (uncorrelated) clock models, under which each branch in the phylogenetic tree has its own molecular substitution rate [3]. Branch rates can be drawn from a range of probability distributions including log-normal, exponential, gamma, and inverse-gamma distributions [3, 17, 18]. This class of models is widely used, and has aided insight into many recent biological problems, including the 2016 Zika virus outbreak [19] and the COVID-19 pandemic [20].

Finally, although not the focus of this article, the class of correlated clock models assumes some form of auto-correlation between rates over time. The correlation itself can invoke a range of stochastic models, including compound Poisson [21] and CIR processes [17], Bayesian parametric models [22], or it can exist as a series of local clocks [23, 24]. While these models may be more biologically realistic than uncorrelated models [17], due to their correlated and discrete natures the time required for MCMC to achieve convergence can be cumbersome, particularly for larger datasets [24]. In the remainder of this paper we only consider uncorrelated relaxed clock models.

With the overwhelming availability of biological sequence data, the development of efficient Bayesian phylogenetic methods is more important than ever. The performance of MCMC is dependent not only on computational performance but also the efficacy of an MCMC setup to achieve convergence. A critical task therein lies the further advancement of MCMC operators. Recent developments in this area include the advancement of guided tree proposals [25–27], coupled MCMC [28, 29], adaptive multivariate transition kernels [30], and other explorative proposal kernels such as the Bactrian and mirror kernels [31, 32]. In the case of relaxed clocks, informed tree proposals can account for correlations between substitution rates and divergence times [33]. The rate parameterisation itself can also affect the ability to "mix" during MCMC [3, 18, 33].

While a range of advanced operators and other MCMC optimisation methods have arisen over the years, there has yet to be a large scale performance benchmarking of such methods as applied to the relaxed clock model. In this article, we systematically evaluate how the relaxed clock model can benefit from i) adaptive operator weighting, ii) substitution rate

parameterisation, iii) Bactrian proposal kernels [31], iv) tree operators which account for correlations between substitution rates and times, and v) adaptive multivariate operators [30]. The discussed methods are implemented in the ORC package and compared using BEAST 2 [10].

## Models and methods

### Preliminaries

Let $\mathcal{T}$ be a binary rooted time tree with $N$ taxa, and data $D$ associated with the tips, such as a multiple sequence alignment with $L$ sites, morphological data, or geographic locations. The posterior density of a phylogenetic model is described by

$$p(\mathcal{T}, \vec{\mathcal{R}}, \sigma, \mu_C, \theta | D) \propto p(D | \mathcal{T}, r(\vec{\mathcal{R}}), \mu_C, \theta) \, p(\mathcal{T} | \theta) \, p(\vec{\mathcal{R}} | \sigma) \, p(\sigma) \, p(\mu_C) \, p(\theta) \qquad (1)$$

where $\sigma$ and $\mu_C$ represent clock model related parameters, and $p(\mathcal{T} | \theta)$ is the tree prior where $\theta$ describes parameters related to the tree branching or coalescent process. The tree likelihood $p(D | \mathcal{T}, r(\vec{\mathcal{R}}), \mu_C, \theta)$ has $\mu_C$ as the overall clock rate and $\vec{\mathcal{R}}$ is an abstracted vector of branch rates which is transformed into real rates by function $r(\vec{\mathcal{R}})$. Branch rates have a mean of 1 under the prior to avoid non-identifiability with the clock rate $\mu_C$. Three methods of representing rates as $\vec{\mathcal{R}}$ are presented in Branch rate parameterisations.

Let $t_i$ be the height (time) of node $i$. Each node $i$ in $\mathcal{T}$, except for the root, is associated with a parental branch length $\tau_i$ (the height difference between $i$ and its parent) and a parental branch substitution rate $r_i = r(\mathcal{R}_i)$. In an uncorrelated relaxed clock model, each of the $2N - 2$ elements in $\vec{\mathcal{R}}$ are independently distributed under the prior $p(\vec{\mathcal{R}} | \sigma)$.

The posterior distribution is sampled by the Metropolis-Hastings-Green MCMC algorithm [7, 8, 34], under which the probability of accepting proposed state $x'$ from state $x$ is equal to:

$$\alpha(x' | x) = \min\left(1, \frac{p(x'|D)}{p(x|D)} \, \frac{q(x|x')}{q(x'|x)} \, |J|\right) \qquad (2)$$

where $q(x'|x)$ is the transition kernel: the probability of proposing state $x'$ from state $x$. The ratio between the two $\frac{q(x|x')}{q(x'|x)}$ is known as the Hastings ratio [8]. The determinant of the Jacobian matrix $|J|$, known as the Green ratio, solves the dimension-matching problem for proposals which operate on multiple terms across one or more spaces [34, 35].

### Branch rate parameterisations

In Bayesian inference, the way parameters are represented in the model can affect the mixing ability of the model and the meaning of the model itself [36]. Three methods for parameterising substitution rates are described below. Each parameterisation is associated with i) an abstraction of the branch rate vector $\vec{\mathcal{R}}$, ii) some function for transforming this parameter into unabstracted branch rates $r(\vec{\mathcal{R}})$, and iii) a prior density function of the abstraction $p(\vec{\mathcal{R}} | \sigma)$. The three methods are summarised in Fig 1.

**1. Real rates.** The natural (and unabstracted) parameterisation of a substitution rate is a real number $\mathcal{R}_i \in \mathbb{R}, \mathcal{R}_i > 0$ which is equal to the rate itself. Under the *real* parameterisation:

$$r(\vec{\mathcal{R}}) = \vec{\mathcal{R}}. \qquad (3)$$

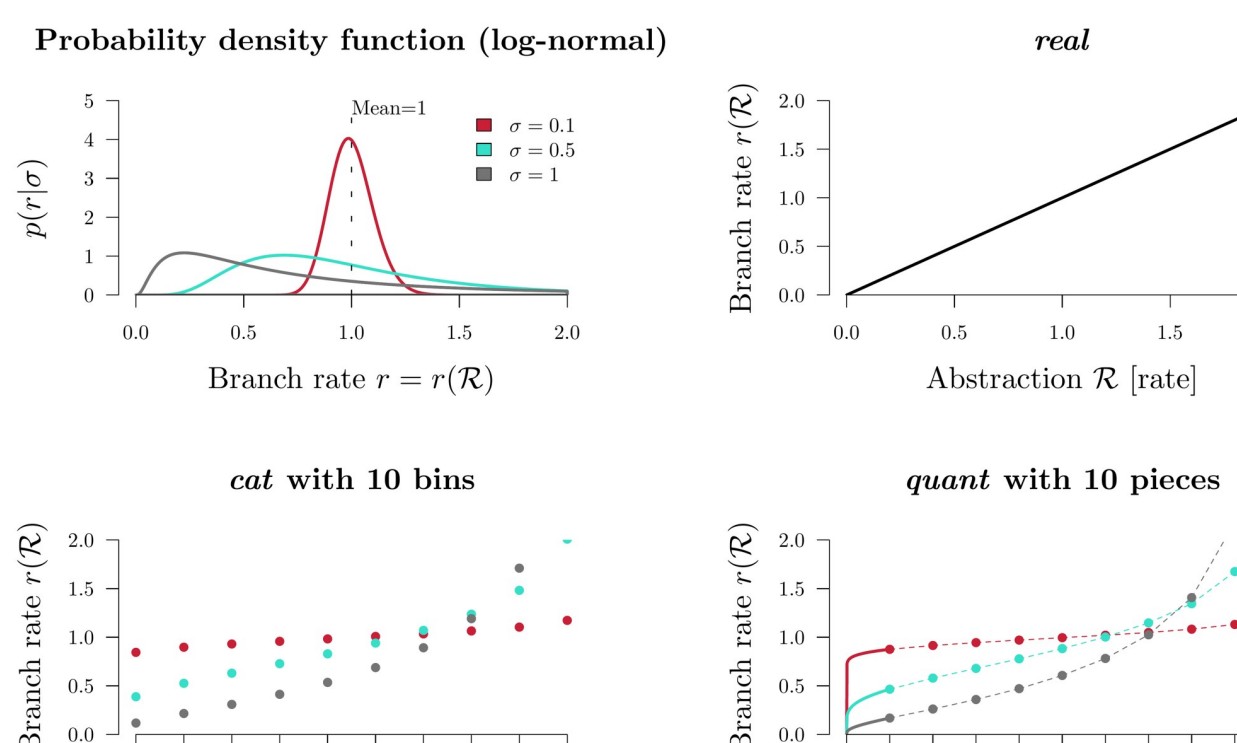

**Fig 1. Branch rate parameterisations.** Top left: the prior density of a branch rate $r$ under a Log-normal($-0.5\sigma^2$, $\sigma$) distribution (with its mean fixed at 1). The function for transforming $\mathcal{R}$ into branch rates $r(\mathcal{R})$ is depicted for *real* (top right), *cat* (bottom left), and *quant* (bottom right). For visualisation purposes, there are only 10 bins/pieces displayed, however in practice we use $2N-2$ bins for *cat* and 100 pieces for *quant*. The first and final *quant* pieces are equal to the underlying function (solid lines) however the pieces in between use linear approximations of this function (dashed lines).

Under a log-normal clock prior $p(\vec{\mathcal{R}}\,|\sigma)$, rates are distributed with a mean of 1:

$$p(\mathcal{R}_i|\sigma) = \frac{1}{\mathcal{R}_i\sigma\sqrt{2\pi}} \exp\left(-\frac{(\ln\mathcal{R}_i - \mu)^2}{2\sigma^2}\right) \tag{4}$$

where $\mu = -0.5\sigma^2$ is set such that the expected value is 1. In this article we only consider log-normal clock priors, however the methods discussed are general.

Zhang and Drummond 2020 introduced a series of tree operators which propose node heights and branch rates, such that the resulting genetic distances ($r_i \times \tau_i$) remain constant [33]. These operators account for correlations between branch rates and branch times. By keeping the genetic distance of each branch constant, the likelihood is unaltered by the proposal.

**2. Categories.** The category parameterisation *cat* is an abstraction of the *real* parameterisation. Each of the $2N-2$ branches are assigned an integer from 0 to $n-1$:

$$\vec{\mathcal{R}} \in \{0, 1, \ldots, n-1\}^{2N-2}. \tag{5}$$

These integers correspond to $n$ rate categories ([Fig 1]). Let $f(x|\sigma)$ be the probability density function (PDF) and let $F(x|\sigma) = \int_0^x f(t|\sigma)\,dt$ be the cumulative distribution function (CDF) of

the prior distribution used by the underlying *real* clock model (a log-normal distribution for the purposes of this article). In the *cat* parameterisation, $f(x|\sigma)$ is discretised into $n$ bins and each element within $\vec{\mathcal{R}}$ points to one such bin. The rate of each bin is equal to its median value:

$$r(\mathcal{R}_i) = F^{-1}(\frac{\mathcal{R}_i + 0.5}{n}), \tag{6}$$

where $F^{-1}$ is the inverse cumulative distribution function (i-CDF). The domain of $\vec{\mathcal{R}}$ is uniformly distributed under the prior:

$$p(\mathcal{R}_i|\sigma) = p(\mathcal{R}_i) = \frac{1}{n}. \tag{7}$$

The key advantage of the *cat* parameterisation is the removal of a term from the posterior density (Eq 1), or more accurately the replacement of a non-trivial $p(\vec{\mathcal{R}}|\sigma)$ term with that of a uniform prior. This may facilitate efficient exploration of the posterior distribution by MCMC.

This parameterisation has been widely used in BEAST and BEAST 2 analyses [3]. However, the recently developed constant distance operators—which are incompatible with the *cat* parameterisation—can yield an increase in mixing rate under *real* by up to an order of magnitude over that of *cat*, depending on the dataset [33].

**3. Quantiles.** Finally, rates can be parameterised as real numbers describing the rate's quantile with respect to some underlying clock model distribution [18]. Under the *quant* parameterisation, each of the $2N - 2$ elements in $\vec{\mathcal{R}}$ are uniformly distributed.

$$\vec{\mathcal{R}} \in \mathbb{R}^{2N-2}, 0 < \mathcal{R}_i < 1 \tag{8}$$

$$p(\mathcal{R}_i|\sigma) = p(\mathcal{R}_i) = 1 \tag{9}$$

Transforming these quantiles into rates invokes the i-CDF of the underlying *real* clock model distribution. Evaluation of the log-normal i-CDF is has high computational costs and therefore an approximation is used instead.

$$r(\mathcal{R}_i) = \hat{F}^{-1}(\mathcal{R}_i) \tag{10}$$

where $\hat{F}^{-1}$ is a linear piecewise approximation with 100 pieces. While this approach has clear similarities with *cat*, the domain of rates here is continuous instead of discrete. In this project we extended the family of constant distance operators [33] so that they are compatible with *quant*. Further details on the *quant* piecewise approximation and constant distance operators can be found in S1 Appendix.

## Clock model operators

The weight of an operator determines the probability of the operator being selected. Weights are typically fixed throughout MCMC. In BEAST 2, operators can have their own tunable parameter *s*, which determines the step size of the operator. This term is tuned over the course of MCMC by achieving a target acceptance rate, typically 0.234 [10, 37, 38]. We define clock model operators as those which generate proposals for either $\vec{\mathcal{R}}$ or $\sigma$. Pre-existing BEAST 2 clock model operators are summarised in Table 1, and further operators are introduced throughout this paper.

**Table 1. Summary of pre-existing BEAST 2 operators.**

| Operator | Description | Parameters | Parameterisations |
|----------|-------------|------------|-------------------|
| RandomWalk | Moves a single element by a tunable amount. | $\vec{\mathcal{R}}, \sigma$ | *cat, real, quant* |
| Scale | Applies RandomWalk on the log-transformation (suitable for parameters with positive domains). | $\vec{\mathcal{R}}, \sigma$ | *cat, real, quant* |
| Interval | Applies RandomWalk on the logit-transformation (suitable for parameters with upper and lower limits). | $\vec{\mathcal{R}}$ | *quant* |
| Swap | Swaps two random elements within the vector [3]. | $\vec{\mathcal{R}}$ | *cat, real, quant* |
| Uniform | Resamples one element in the vector from a uniform distribution. | $\vec{\mathcal{R}}$ | *cat, quant* |
| ConstantDistance | Adjusts an internal node height and recalculates all incident branch rates such that the genetic distances remain constant [33]. | $\vec{\mathcal{R}}, t$ | *real, quant* |
| SimpleDistance | Applies ConstantDistance to the root node [33]. | $\vec{\mathcal{R}}, t$ | *real, quant* |
| SmallPulley | Proposes new branch rates incident to the root such that their combined genetic distance is constant [33]. | $\vec{\mathcal{R}}$ | *real, quant* |
| CisScale | Applies Scale to σ. Then recomputes all rates such that their quantiles are constant (for *real* [33]) or recomputes all quantiles such that their rates are constant (*quant*). | $\vec{\mathcal{R}}, \sigma$ | *real, quant* |

Summary of pre-existing BEAST 2 operators, which apply to either branch rates $\vec{\mathcal{R}}$ or the clock standard deviation σ, and the substitution rate parameterisation they apply to. ConstantDistance and SimpleDistance also adjust node heights *t*.

The family of constant distance operators (ConstantDistance, SimpleDistance, and SmallPulley [33]) are best suited for larger datasets (or datasets with strong signal) where the likelihood distribution is peaked (Fig 1). While simple one dimensional operators such as RandomWalk or Scale must take small steps in order to stay "on the ridge" of the likelihood function, the constant distance operators "wander along the ridge" by ensuring that genetic distances are constant after the proposal.

Scale and CisScale both operate on the clock model standard deviation σ however they behave differently in the *real* and *quant* parameterisations (Fig 2). In *real*, large proposals of $\sigma \to \sigma'$ made by Scale could incur large penalties in the clock model prior density $p(\vec{\mathcal{R}}|\sigma')$ and thus may be rejected quite often. This led to the development of the fast clock scaler [33] (herein referred to as CisScale). This operator recomputes all branch rates $\vec{\mathcal{R}} \to \vec{\mathcal{R}}'$ such that their quantiles under the new clock model prior remain constant $p(\vec{\mathcal{R}}|\sigma) = p(\vec{\mathcal{R}}'|\sigma')$. In contrast, a proposal made by Scale $\sigma \to \sigma'$ under the *quant* parameterisation implicitly alters all branch rates $r(\vec{\mathcal{R}})$ while leaving the quantiles $\vec{\mathcal{R}}$ themselves constant. Whereas, application of CisScale under *quant* results in all quantiles being recomputed $\vec{\mathcal{R}} \to \vec{\mathcal{R}}'$ such that their rates are constant, i.e. $r(\vec{\mathcal{R}}) = r(\vec{\mathcal{R}}')$. In summary, Scale and CisScale propose rates/quantiles in the opposite (trans) or same (cis) space that the clock model is parameterised under (Fig 2).

## Adaptive operator weighting

It is not always clear which operator weighting scheme is best for a given dataset. In this article we introduce AdaptiveOperatorSampler—a meta-operator which learns the weights of other operators during MCMC and then samples these operators according to their learned weights. This meta-operator undergoes three phases. In the first phase (burn-in), AdaptiveOperatorSampler samples from its set of sub-operators uniformly at random. In the second phase (learn-in), the meta-operator starts learning several terms detailed below whilst continuing to sample operators uniformly at random. In its final phase,

## $\sigma$ scale operators

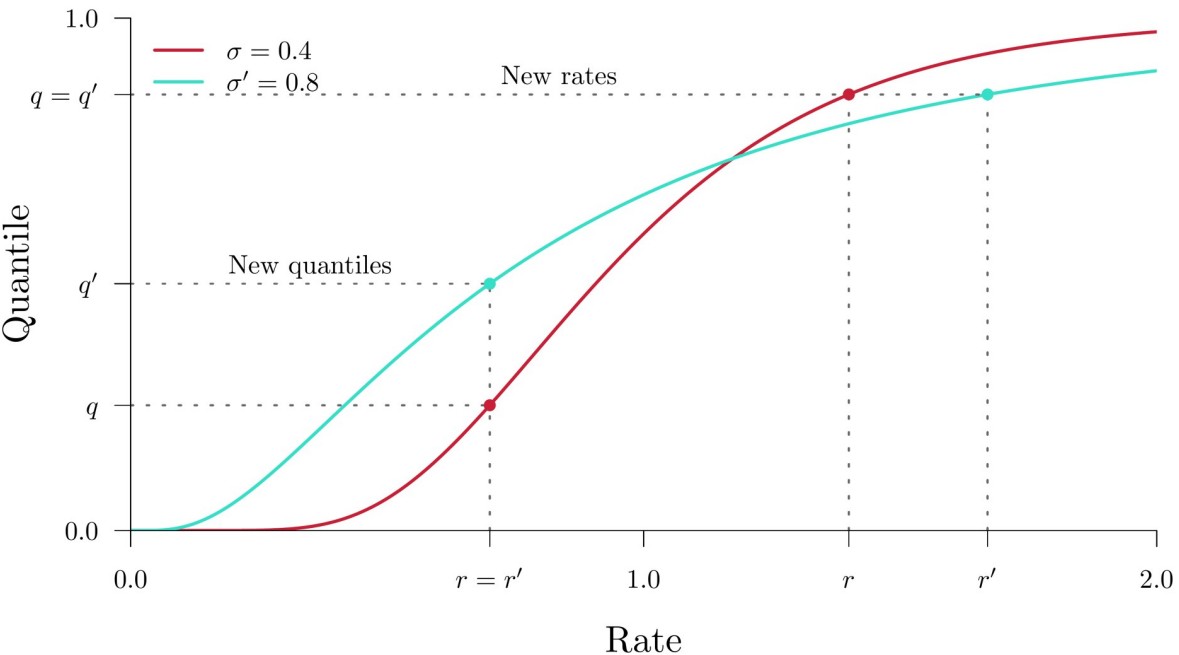

**Fig 2. Clock standard deviation scale operators.** The two operators above propose a clock standard deviation $\sigma \to \sigma'$. Then, either the new quantiles are such that the rates remain constant ("New quantiles", above) or the new rates are such that the quantiles remain constant ("New rates"). In the *real* parameterisation, these two operators are known as `Scale` and `CisScale`, respectively. Whereas, in *quant*, they are known as `CisScale` and `Scale`.

`AdaptiveOperatorSampler` samples operators (denoted by $\omega$) using the following distribution:

$$p(\omega) \propto \begin{cases} 1 & \text{with probability } \Omega \\[2ex] \dfrac{1}{\mathbb{T}(\omega)} \displaystyle\sum_{p \in \text{POI}} \sum_{x \in \text{accepts}(\omega)} \mathbb{D}(x_p, x'_p) & \text{with probability } 1 - \Omega \end{cases} \tag{11}$$

where $\mathbb{T}(\omega)$ is the cumulative computational time spent on each operator, $\mathbb{D}$ is a distance function, and we use $\Omega = 0.01$ to allow any sub-operator to be sampled regardless of its performance. The parameters of interest (POI) may be either a set of numerical parameters (such as branch rates or node heights), or it may be the tree itself, but it cannot be both in its current form. The distance between state $x_p$ and its (accepted) proposal $x'_p$ with respect to parameter $p$ is determined by

$$\mathbb{D}(x_p, x'_p) = \begin{cases} \text{RF}(x_p, x'_p)^2 & \text{if } p \text{ is a tree} \\[2ex] \dfrac{1}{|p|} \left[ \dfrac{\|x_p - x'_p\|}{\sigma_p} \right]^2 & \text{if } p \text{ is numerical} \end{cases} \tag{12}$$

where $\text{RF}$ is the Robinson-Foulds tree distance [39], and $|p|$ is the number of dimensions of numerical parameter $p$ (1 for $\sigma$, $2N - 2$ for $\vec{\mathcal{R}}$, and $2N - 1$ for node heights $t$). The remaining terms are trained during the second and third phases: the sample standard deviation $\sigma_p$ of each

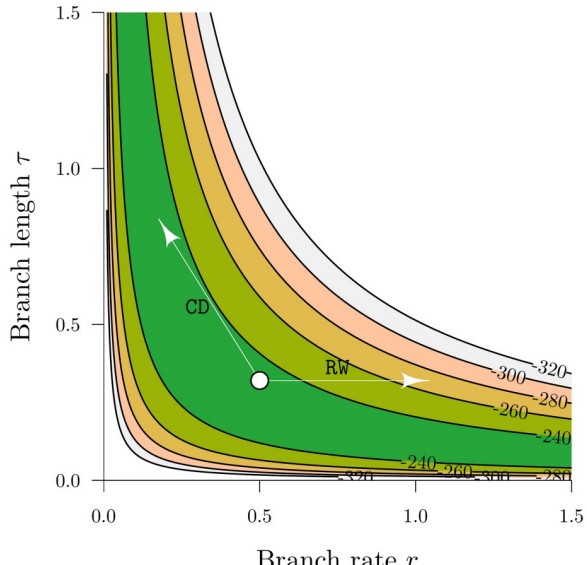

**Fig 3. Traversing likelihood space.** The z-axes above are the log-likelihoods of the genetic distance $r \times \tau$ between two simulated nucleic acid sequences of length $L$, under the Jukes-Cantor substitution model [40]. Two possible proposals from the current state (white circle) are depicted. These proposals are generated by the `RandomWalk` (RW) and `ConstantDistance` (CD) operators. In the low signal dataset ($L = 0.1$kb), both operators can traverse the likelihood space effectively. However, the exact same proposal by `RandomWalk` incurs a much larger likelihood penalty in the $L = 0.5$kb dataset by "falling off the ridge", in contrast to `ConstantDistance` which "walks along the ridge". This discrepancy is even stronger for larger datasets and thus necessitates the use of operators such as `ConstantDistance` which account for correlations between branch lengths and rates.

numerical parameter of interest $p$, the cumulative computational runtime spent on each operator $\mathbb{T}(\omega)$, and the summed distances $\sum_x \mathbb{D}(x_p, x'_p)$.

Under Eqs 11 and 12, operators which effect larger changes on the parameters of interest, in shorter runtime, are sampled with greater probabilities. Division of the squared distance by a parameter's sample variance $\sigma_p^2$ enables comparison between numerical parameters which exist in different spaces.

Datasets which contain very poor signal (or small $L$) are likely to mix better when more weight is placed on bold operators (Fig 3). We therefore introduce the `SampleFromPrior` $(\vec{x})$ operator. This operator resamples $\psi$ randomly selected elements within vector $\vec{x}$ from their prior distributions, where $\psi \sim \text{Binomial}\left(n = |\vec{x}|, p = \frac{s}{|\vec{x}|}\right)$ for tunable term $s$. `Sample-FromPrior` is included among the set of operators under `AdaptiveOperatorSampler` and serves to make the boldest proposals for datasets with poor signal.

In this article we apply three instances of the `AdaptiveOperatorSampler` meta-operator to the *real*, *cat*, and *quant* parameterisations. These are summarised in Table 2.

## Bactrian proposal kernel

The step size of a proposal kernel $q(x'|x)$ should be such that the proposed state $x'$ is sufficiently far from the current state $x$ to explore vast areas of parameter space, but not so large that the proposal is rejected too often [37]. Operators which attain an acceptance probability of 0.234 are often considered to have arrived at a suitable midpoint between these two extremes [10, 37]. The standard uniform distribution kernel has recently been challenged by the family of

**Table 2. Summary of `AdaptiveOperatorSampler` operators and their parameters of interest (POI).**

| Meta-operator | POI | Operators |
|---|---|---|
| `AdaptiveOperatorSampler(`$\sigma$`)` | $\sigma$ | `CisScale(`$\sigma, \vec{\mathcal{R}}$`)` |
| | | `RandomWalk(`$\sigma$`)` |
| | | `Scale(`$\sigma$`)` |
| | | `SampleFromPrior(`$\sigma$`)` |
| `AdaptiveOperatorSampler(`$\vec{\mathcal{R}}$`)` | $\vec{\mathcal{R}}, t$ | `ConstantDistance(`$\vec{\mathcal{R}}, t$`)` |
| | | `RandomWalk(`$\vec{\mathcal{R}}$`)` |
| | | `Scale(`$\vec{\mathcal{R}}$`)` |
| | | `Interval(`$\vec{\mathcal{R}}$`)` |
| | | `Swap(`$\vec{\mathcal{R}}$`)` |
| | | `SampleFromPrior(`$\vec{\mathcal{R}}$`)` |
| `AdaptiveOperatorSampler(root)` | $\vec{\mathcal{R}}, t$ | `SimpleDistance(`$\vec{\mathcal{R}}, t$`)` |
| | | `SmallPulley(`$\vec{\mathcal{R}}, t$`)` |
| `AdaptiveOperatorSampler(leaf)` | $\vec{\mathcal{R}}_{\text{leaf}}, t$ | `ConstantDistance(`$\vec{\mathcal{R}}_{\text{leaf}}, t$`)` |
| | | `LeafAVMVN(`$\vec{\mathcal{R}}_{\text{leaf}}$`)` |
| | | `RandomWalk(`$\vec{\mathcal{R}}_{\text{leaf}}$`)` |
| | | `Scale(`$\vec{\mathcal{R}}_{\text{leaf}}$`)` |
| | | `Interval(`$\vec{\mathcal{R}}_{\text{leaf}}$`)` |
| | | `Swap(`$\vec{\mathcal{R}}_{\text{leaf}}$`)` |
| | | `SampleFromPrior(`$\vec{\mathcal{R}}_{\text{leaf}}$`)` |
| `AdaptiveOperatorSampler(internal)` | $\vec{\mathcal{R}}_{\text{int}}, t$ | `ConstantDistance(`$\vec{\mathcal{R}}_{\text{int}}, t$`)` |
| | | `RandomWalk(`$\vec{\mathcal{R}}_{\text{int}}$`)` |
| | | `Scale(`$\vec{\mathcal{R}}_{\text{int}}$`)` |
| | | `Interval(`$\vec{\mathcal{R}}_{\text{int}}$`)` |
| | | `Swap(`$\vec{\mathcal{R}}_{\text{int}}$`)` |
| | | `SampleFromPrior(`$\vec{\mathcal{R}}_{\text{int}}$`)` |
| `AdaptiveOperatorSampler(NER)` | $\mathcal{T}$ | `NER{}` |
| | | `NER{`$\mathcal{D}_{AE}, \mathcal{D}_{BE}, \mathcal{D}_{CE}$`}` |

Different operators are applicable to different substitution rate parameterisations (Table 1). Nodes are broken down into regions to enable operators to be weighted according to their dimension. `AdaptiveOperatorSampler(root)` applies the root-targeting constant distance operators only [33] while `AdaptiveOperatorSampler(`$\vec{\mathcal{R}}$`)` targets all rates and all nodes heights $t$. Nodes are further broken down into leaf rate $\vec{\mathcal{R}}_{\text{leaf}}$ and internal node rate $\vec{\mathcal{R}}_{\text{int}}$ operators. This facilitates suitable weighting of the `LeafAVMVN` operator, which is only applicable to leaf nodes (4. Screening operators for acceptance rate using simulated data.). In this setup, the `RandomWalk(`$x$`)`, `Scale(`$x$`)`, and `SampleFromPrior(`$x$`)` operators apply to the corresponding set of branch rates $x$, whereas `ConstantDistance(`$x$`, `$t$`)` is only applicable to internal nodes which have at least one child of type $x \in \{\vec{\mathcal{R}}_{\text{leaf}}, \vec{\mathcal{R}}_{\text{int}}\}$. Finally, the Robinson-Foulds distance between trees before and after every proposal accept is used to train the weights behind `AdaptiveOperatorSampler(NER)` (see Narrow exchange rate). In the special case of NER proposals, the `RF` distance is always equal to 1.

Bactrian kernels [31, 32]. The (Gaussian) Bactrian($m$) distribution is defined as the sum of two normal distributions:

$$\Sigma \sim \text{Bactrian}(m) \equiv \frac{1}{2}\text{Normal}(-m, 1 - m^2) + \frac{1}{2}\text{Normal}(m, 1 - m^2) \qquad (13)$$

where $0 \leq m < 1$ describes modality. When $m = 0$, the Bactrian distribution is equivalent to Normal(0, 1). As $m$ approaches 1, the distribution becomes increasingly bimodal (Fig 4). Yang

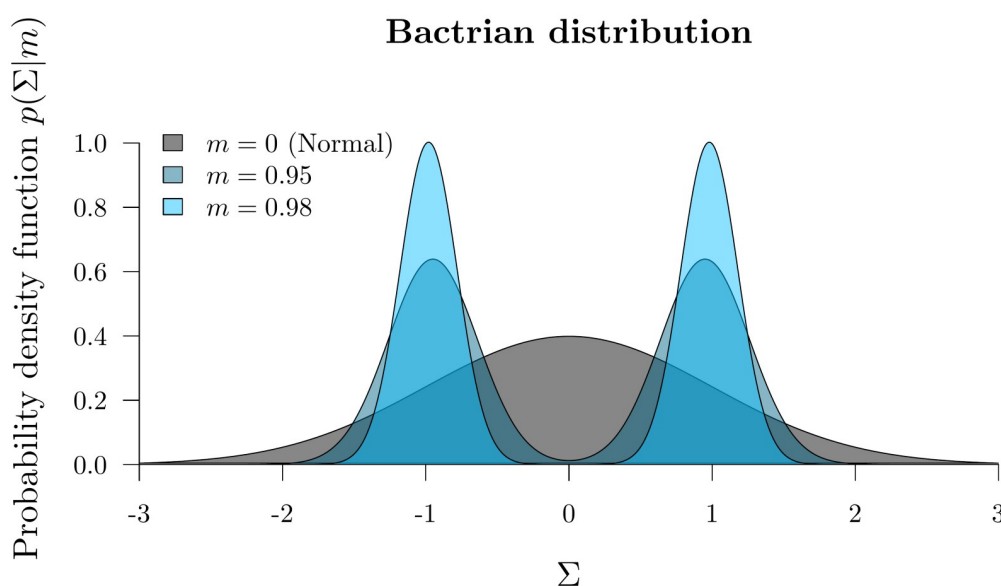

**Fig 4. The Bactrian proposal kernel.** The step size made under a Bactrian proposal kernel is equal to $s\Sigma$ where $\Sigma$ is drawn from the above distribution and $s$ is tunable.

et al. 2013 [31] demonstrate that Bactrian($m = 0.95$) yields a proposal kernel which traverses the posterior distribution more efficiently than the standard uniform kernel, by placing minimal probability on steps which are too small or too large. In this case, a target acceptance probability of around 0.3 is optimal.

In this article we compare the abilities of uniform and Bactrian(0.95) proposal kernels at estimating clock model parameters. The clock model operators which these proposal kernels apply to are described in S1 Appendix.

## Narrow exchange rate

The `NarrowExchange` operator [41], used widely in BEAST [9, 42] and BEAST 2 [10], is similar to nearest neighbour interchange [43], and works as follows (Fig 5):

*Step 1*. Sample an internal/root node $E$ from tree $\mathcal{T}$, where $E$ has grandchildren.

*Step 2*. Identify the child of $E$ with the greater height. Denote this child as $D$ and its sibling as $C$ (i.e. $t_D > t_C$). If $D$ is a leaf node, then reject the proposal.

*Step 3*. Randomly identify the two children of $D$ as $A$ and $B$.

*Step 4*. Relocate the $B - D$ branch onto the $C - E$ branch, so that $B$ and $C$ become siblings and their parent is $D$. All node heights remain constant.

We hypothesised that if `NarrowExchange` was adapted to the relaxed clock model by ensuring that genetic distances remain constant after the proposal (analogous to constant distance operators [33]), then its ability to traverse the state space may improve.

Here, we present the `NarrowExchangeRate` (NER) operator. Let $r_A$, $r_B$, $r_C$, and $r_D$ be the substitution rates of nodes $A$, $B$, $C$, and $D$, respectively. In addition to the modest topological change applied by `NarrowExchange`, NER also proposes new branch rates $r_A{}'$, $r_B{}'$, $r_C{}'$, and $r_D{}'$. While NER does not alter $t_D$ (i.e. $t_D{}' \leftarrow t_D$), we also consider NERw—a special case of

**NarrowExchange**          $\mathcal{T} \longrightarrow \mathcal{T}`$

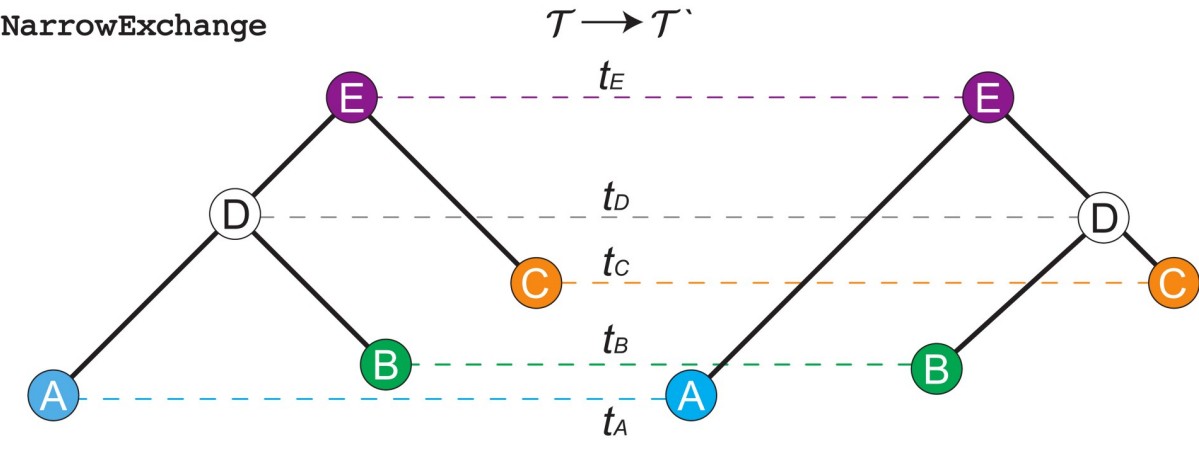

**NarrowExchangeRate**          $\mathcal{T} \longrightarrow \mathcal{T}`$

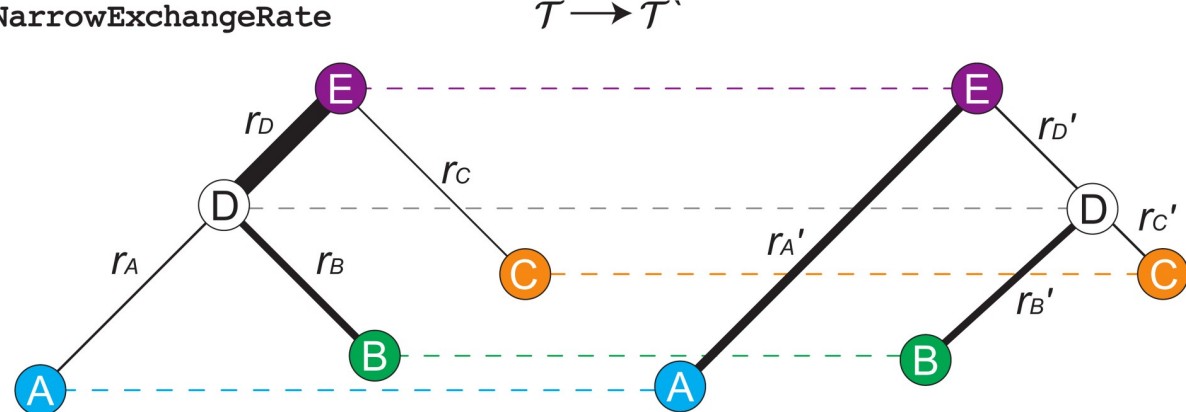

**Fig 5. Depiction of `NarrowExchange` and `NarrowExchangeRate` operators.** Proposals are denoted by $\mathcal{T} \to \mathcal{T}'$. The vertical axes correspond to node heights $t$. In the bottom figure, branch rates $r$ are indicated by line width and therefore genetic distances are equal to the width of each branch multiplied by its length. In this example, the $\mathcal{D}_{AE}$ and $\mathcal{D}_{CE}$ constraints are satisfied.

the NER operator which embarks $t_D$ on a random walk:

$$t_D' \leftarrow t_D + s\Sigma \tag{14}$$

for random walk step size $s\Sigma$ where $s$ is tunable and $\Sigma$ is drawn from a uniform or Bactrian distribution. NER (and NERw) are compatible with both the *real* and *quant* parameterisations. Analogous to the `ConstantDistance` operator, the proposed rates ensure that the genetic distances between nodes $A$, $B$, $C$, and $E$ are constant. Let $\mathcal{D}_{ij}$ be the constraint defined by a constant genetic distance between nodes $i$ and $j$ before and after the proposal. There are six pairwise distances between these four nodes and therefore there are six such constraints:

$$\mathcal{D}_{AB}: \quad \begin{aligned} &r_A(t_D - t_A) + r_B(t_D - t_B) = \\ &r_A'(t_E - t_A) + r_D'(t_E - t_D') + r_B'(t_D' - t_B) \end{aligned} \tag{15}$$

$$\mathcal{D}_{AC}: \quad \begin{aligned} &r_A(t_D - t_A) + r_D(t_E - t_D) + r_C(t_E - t_C) = \\ &r_A'(t_E - t_A) + r_D'(t_E - t_D') + r_C'(t_D' - t_C) \end{aligned} \tag{16}$$

$$\mathcal{D}_{AE}: \qquad r_A(t_D - t_A) + r_D(t_E - t_D) = \\ r_A'(t_E - t_A) \tag{17}$$

$$\mathcal{D}_{BC}: \qquad r_B(t_D - t_B) + r_D(t_E - t_D) + r_C(t_E - t_D) = \\ r_B'(t_D' - t_B) + r_C'(t_D' - t_C) \tag{18}$$

$$\mathcal{D}_{BE}: \qquad r_B(t_D - t_B) + r_D(t_E - t_D) = \\ r_B'(t_D' - t_B) + r_D'(t_E - t_D') \tag{19}$$

$$\mathcal{D}_{CE}: \qquad r_C(t_E - t_C) = \\ r_C'(t_D' - t_C) + r_D'(t_E - t_D') \tag{20}$$

Further constraints are imposed by the model itself:

$$r_i, r_i' > 0 \text{ for } i \in \{A, B, C, D\} \tag{21}$$

$$t_B, t_C < t_D' < t_E. \tag{22}$$

Unfortunately, there is no solution to all six $\mathcal{D}_{ij}$ constraints unless non-positive rates or illegal trees are permitted. Therefore instead of conserving all six pairwise distances, NER conserves a subset of distances. It is not immediately clear which subset should be conserved.

**Automated generation of operators and constraint satisfaction.** The total space of NER operators consists of all possible subsets of distance constraints (i.e. $\{\}, \{\mathcal{D}_{AB}\}, \{\mathcal{D}_{AC}\}, \ldots, \{\mathcal{D}_{AB}, \mathcal{D}_{AC}, \mathcal{D}_{AE}, \mathcal{D}_{BC}, \mathcal{D}_{BE}, \mathcal{D}_{CE}\}$) that are solvable. The simplest NER kernel—the null operator denoted by NER{}—does not satisfy any distance constraints and is equivalent to `NarrowExchange`. To determine which NER variants have the best performance, we developed an automated pipeline for generating and testing these operators.

**1. Solution finding.** Using standard analytical linear-system solving libraries in MATLAB [44], the $2^6 = 64$ subsets of distance constraints were solved. 54 of the 64 subsets were found to be solvable, and the unsolvables were discarded.

**2. Solving Jacobian determinants.** The determinant of the Jacobian matrix $J$ is required for computing the Green ratio of this proposal. $J$ is defined as

$$J = \begin{bmatrix} \frac{\partial r_A'}{\partial r_A} & \frac{\partial r_A'}{\partial r_B} & \frac{\partial r_A'}{\partial r_C} & \frac{\partial r_A'}{\partial r_D} & \frac{\partial r_A'}{\partial t_D} \\ \frac{\partial r_B'}{\partial r_A} & \frac{\partial r_B'}{\partial r_B} & \frac{\partial r_B'}{\partial r_C} & \frac{\partial r_B'}{\partial r_D} & \frac{\partial r_B'}{\partial t_D} \\ \frac{\partial r_C'}{\partial r_A} & \frac{\partial r_C'}{\partial r_B} & \frac{\partial r_C'}{\partial r_C} & \frac{\partial r_C'}{\partial r_D} & \frac{\partial r_C'}{\partial t_D} \\ \frac{\partial r_D'}{\partial r_A} & \frac{\partial r_D'}{\partial r_B} & \frac{\partial r_D'}{\partial r_C} & \frac{\partial r_D'}{\partial r_D} & \frac{\partial r_D'}{\partial t_D} \\ \frac{\partial t_D'}{\partial r_A} & \frac{\partial t_D'}{\partial r_B} & \frac{\partial t_D'}{\partial r_C} & \frac{\partial t_D'}{\partial r_D} & \frac{\partial t_D'}{\partial t_D} \end{bmatrix}. \tag{23}$$

Solving the determinant $|J|$ invokes standard analytical differentiation and linear algebra libraries of MATLAB. 6 of the 54 solvable operators were found to have $|J| = 0$, corresponding to irreversible proposals, and were discarded.

**3. Automated generation of BEAST 2 operators.**    Java class files were generated using string processing. Each class corresponded to a single operator, extended the class of a meta-NER-operator, and consisted of the solutions found in **1** and the Jacobian determinant found in **2**. |*J*| is further augmented if the *quant* parameterisation is employed (S1 Appendix). One such operator is expressed in **Algorithm 1** and a second in S1 Appendix.

**Algorithm 1** The NER$\{\mathcal{D}_{AE}, \mathcal{D}_{BE}, \mathcal{D}_{CE}\}$ operator.

```
 1: procedure PROPOSAL(t_A, t_B, t_C, t_D, t_E, r_A, r_B, r_C, r_D)
 2:
 3: sΣ ← getRandomWalkSize()      ▷Random walk size is 0 unless this is
NERw
 4: t'_D ← t_D + sΣ                 ▷ Propose new node height for D
 5:
```

6: $r'_A \leftarrow \frac{r_A(t_D - t_A) + r_D(t_E - t_D)}{t_E - t_A}$ ▷Propose new rates

7: $r'_B \leftarrow \frac{r_B(t_D - t_B) + r_D(t'_D - t_D)}{t'_D - t_B}$

8: $r'_C \leftarrow \frac{r_C(t_E - t_C) - r_D(t_E - t'_D)}{t'_D - t_C}$

9: $r'_D \leftarrow r_D$

```
10:
```

11: $|J| \leftarrow \frac{(t_D - t_A)(t_D - t_B)(t_E - t_C)}{(t_E - t_A)(t'_D - t_B)(t'_D - t_C)}$ ▷Calculate Jacobian determinant

```
12: return (r'_A, r'_B, r'_C, r'_D, t'_D, |J|)
```

**4. Screening operators for acceptance rate using simulated data.**    Selecting the best NER variant to proceed to benchmarking on empirical data (Results) was determined by performing MCMC on simulated data, measuring the acceptance rates of each of the 96 NER/NERw variants, and comparing them with the null operator NER{} / NarrowExchange. In total, there were 300 simulated datasets each with $N = 30$ taxa and varying alignment lengths.

These experiments showed that NER variants which satisfied the genetic distances between nodes *B* and *A* (i.e. $\mathcal{D}_{AB}$) or between *B* and *C* (i.e. $\mathcal{D}_{BC}$) usually performed worse than the standard NarrowExchange operator (Fig 6). This is an intuitive result. If there is high uncertainty in the positioning of *B* with respect to *A* and *C*, then there is no value in respecting either of these distance constraints, and the proposals made to the rates may often be too extreme or the Green ratio |*J*| too small for the proposal to be accepted.

Fig 6 also revealed a cluster of NER variants which—under the conditions of the simulation—performed better than the null operator NER{} around 25% of the time and performed worse around 10% of the time. One such operator was NER$\{\mathcal{D}_{AE}, \mathcal{D}_{BE}, \mathcal{D}_{CE}\}$ (**Algorithm 1**). This variant conserves the genetic distance between nodes *A*, *B*, *C* and their grandparent *E*. This operator performed well when branch rates had a large variance ($\sigma > 0.5$), corresponding to non clock-like data. On the other hand, the null operator NER{} performed better on shorter sequences ($L < 1\text{kb}$) with weaker signal. Overall, NER$\{\mathcal{D}_{AE}, \mathcal{D}_{BE}, \mathcal{D}_{CE}\}$ outperformed the standard NarrowExchange operator when the data was not clock-like and contained sufficient signal.

Finally, this initial screening showed that applying a (Bactrian) random walk to the node height $t_D$ made the operator worse. This effect was most dominant for the NER variants which satisfied distance constraints (i.e. the operators which are not NER{}).

Although there were several operators which behaved equivalently during this initial screening process, we selected NER$\{\mathcal{D}_{AE}, \mathcal{D}_{BE}, \mathcal{D}_{CE}\}$ to proceed to benchmarking (Results). Due to the apparent sensitivity of NER operators to the data, we introduce the adaptive operator AdaptiveOperatorSampler(NER) which allows the operator scheme to fall back on the standard NarrowExchange in the event of NER$\{\mathcal{D}_{AE}, \mathcal{D}_{BE}, \mathcal{D}_{CE}\}$ performing poorly (Table 2).

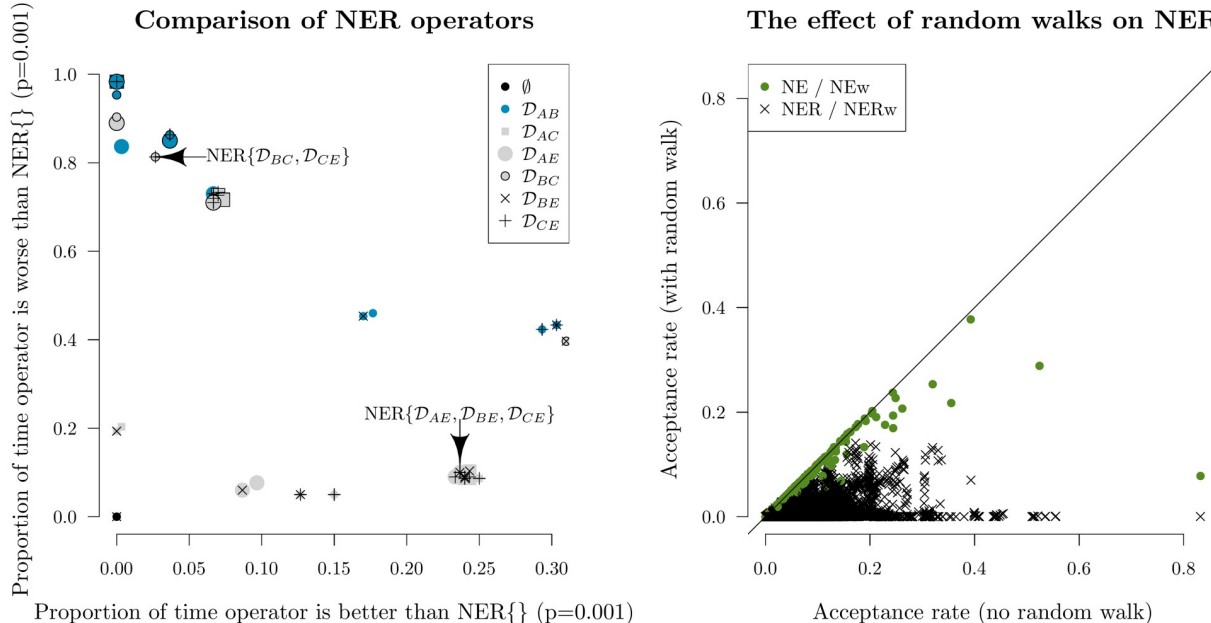

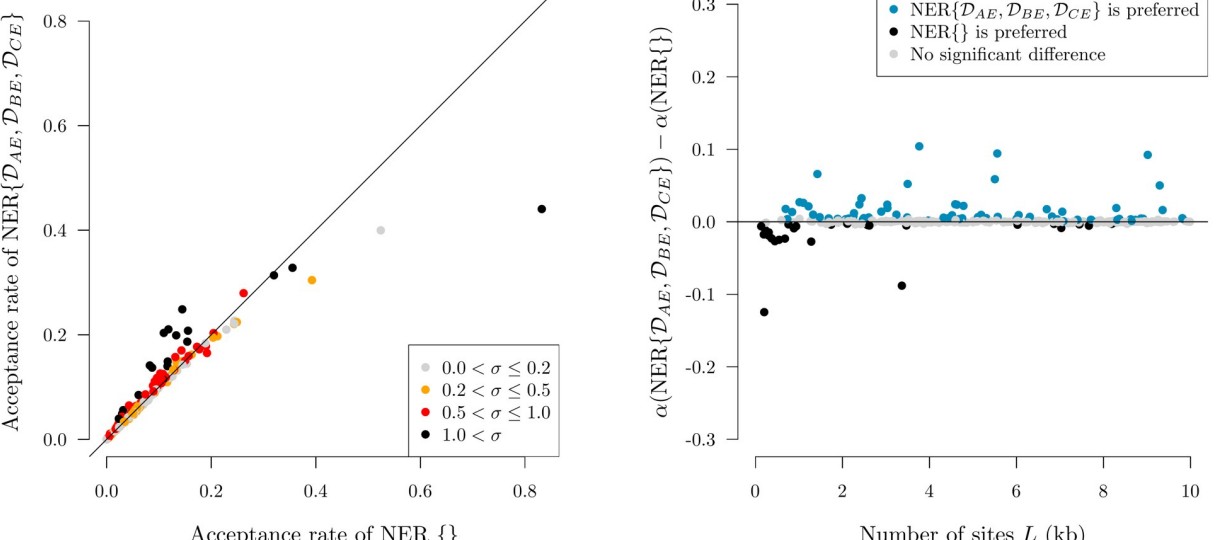

**Fig 6. Screening of NER and NERw variants by acceptance rate.** Top left: comparison of NER variants with the null operator NER{} = `NarrowExchange`. Each operator is represented by a single point, uniquely encoded by the point stylings. The number of times each operator is proposed and accepted is compared with that of NER{}, and one-sided z-tests are performed to assess the statistical significance between the two acceptance rates ($p = 0.001$). This process is repeated across 300 simulated datasets. The axes of each plot are the proportion of these 300 simulations for which there is evidence that the operator is significantly better than NER{} (x-axis) or worse than NER{} (y-axis). Top right: comparison of NER and NERw acceptance rates. Each point is one NER/NERw variant from a single simulation. Bottom: relationship between the acceptance rates $\alpha$ of NER{$\mathcal{D}_{AE}, \mathcal{D}_{BE}, \mathcal{D}_{CE}$} and NER{} with the clock model standard deviation $\sigma$ and the number of sites $L$. Each point is a single simulation.

## An adaptive leaf rate operator

The adaptable variance multivariate normal (AVMVN) kernel learns correlations between parameters during MCMC [30, 42]. Baele et al. 2017 observed a large increase ($\approx 5 - 10\times$) in sampling efficiency from using the AVMVN kernel substitution model parameters [30]. Here,

we consider application of the AVMVN kernel to the branch rates of leaf nodes. This operator, referred to as `LeafAVMVN`, is not readily applicable to internal node branch rates due to their dependencies on tree topology.

**Leaf rate AVMVN kernel.** The AVMVN kernel assumes its parameters live in $x \in \mathbb{R}^N$ for taxon count $N$ and that these parameters follow a multivariate normal distribution with covariance matrix $\Sigma_N$. Hence, the kernel operates on the logarithmic or logistic transformation of the $N$ leaf branch rates, depending on the rate parameterisation:

$$x = \begin{cases} \log r & \text{for } \textit{real} \\ \log \frac{q}{1-q} & \text{for } \textit{quant} \end{cases} \tag{24}$$

where $r$ is a real rate and $q$ is a rate quantile. The AVMVN probability density is defined by

$$\mathcal{AVMVN}(\vec{x}) = \mathcal{MVN}\left(\vec{x}, (1-\beta)\frac{\Sigma_N}{N} + \beta\frac{\mathbb{I}_N}{N}\right), \tag{25}$$

where $\mathcal{MVN}$ is the multivariate normal probability density. $\beta = 0.05$ is a constant which determines the fraction of the proposal determined by the identity matrix $\mathbb{I}_N$, as opposed to the covariance matrix $\Sigma_D$ which is trained during MCMC. Our BEAST 2 implementation of the AVMVN kernel is adapted from that of BEAST [42].

`LeafAVMVN` has the advantage of operating on all $N$ leaf rates simultaneously (as well as learning their correlations), as opposed to `ConstantDistance` which operates on at most 2, or `Scale` which operates on at most 1 leaf rate at a time. As the size of the covariance matrix $\Sigma_N$ grows with the number of taxa $N$, `LeafAVMVN` is likely to be less efficient with larger taxon sets. Therefore, the weight behind this operator is learned by `AdaptiveOperatorSampler`.

To prevent the learned weight behind `LeafAVMVN` from dominating the `AdaptiveOperatorSampler` weighting scheme and therefore inhibiting the mixing of internal node rates, we introduce the `AdaptiveOperatorSampler(leaf)` and `AdaptiveOperatorSampler(internal)` meta-operators which operate exclusively on leaf node rates $\vec{\mathcal{R}}_{\text{leaf}}$ and internal node rates $\vec{\mathcal{R}}_{\text{int}}$ respectively (Table 2). The former employs the `LeafAVMVN` operator and learns its weight during MCMC (after providing it sufficient time to learn $\Sigma_N$).

## Model specification and MCMC settings

In all phylogenetic analyses presented here, we use a Yule [45] tree prior $p(\mathcal{T}|\lambda)$ with birth rate $\lambda \sim$ Log-normal(1, 1.25). Here and throughout the article, a Log-normal($a$, $b$) distribution is parameterised such that $a$ and $b$ are the mean and standard deviation in log-space. The clock standard deviation has a $\sigma \sim$ Gamma(0.5396, 0.3819) prior. Datasets are partitioned into subsequences, where each partition is associated with a distinct HKY substitution model [46]. The transition-transversion ratio $\kappa \sim$ Log-normal(1, 1.25), the four nucleotide frequencies ($f_A, f_C, f_G, f_T$) ~ Dirichlet(10, 10, 10, 10), and the relative clock rate $\mu_C \sim$ Log-normal(−0.18, 0.6) are estimated independently for each partition. The operator scheme ensures that the clock rates $\mu_C$ have a mean of 1 across all partitions. This avoids non-identifiability with branch substitution rates. To enable rapid benchmarking of larger datasets we use BEAGLE for high-performance tree likelihood calculations [47] and coupled MCMC with four chains for efficient mixing [29]. The neighbour joining tree [48] is used as the initial state in each MCMC chain.

Throughout the article, we have introduced four new operators. These are summarised in Table 3.

**Table 3. Summary of clock model operators introduced throughout this article.**

| Operator | Description | Parameters |
|---|---|---|
| `AdaptiveOperatorSampler` | Samples sub-operators proportionally to their weights, which are learned (see Adaptive operator weighting). | $\vec{\mathcal{R}}, \sigma, t\mathcal{T}$ |
| `SampleFromPrior` | Resamples a random number of elements from their prior (see Adaptive operator weighting). | $\vec{\mathcal{R}}, \sigma$ |
| `NarrowExchangeRate` | Moves a branch and recomputes branch rates so that their genetic distances are constant (see Narrow exchange rate). | $\vec{\mathcal{R}}, \mathcal{T}$ |
| `LeafAVMVN` | Proposes new rates for all leaves in one move (see An adaptive leaf rate operator) [30]. | $\vec{\mathcal{R}}$ |

Pre-existing clock model operators are summarised in Table 1. $t$ denotes node heights while $\mathcal{T}$ denotes the whole tree, including topology.

In Table 4, we define all operator configurations which are benchmarked throughout Results.

## Results

To avoid an intractably large search space, the five targets for clock model improvement were evaluated sequentially in the following order: Adaptive operator weighting, Branch rate parameterisations, Bactrian proposal kernel, Narrow exchange rate, and An adaptive leaf rate operator. The four operators introduced in these sections are summarised in Table 3. The setting which was considered to be the best in each step was then incorporated into the following step. This protocol and its outcomes are summarised in Fig 7.

Methodologies were assessed according to the following criteria.

1. **Validation**. This was assessed by measuring the coverage of all estimated parameters in well-calibrated simulation studies. These are presented in S2 Appendix and give confidence operators are implemented correctly.

2. **Mixing of parameters**. Key parameters were evaluated for the number of effective samples generated per hour (ESS/hr). These key parameters were the likelihood $\mathcal{L}$ and prior $p$ densities, tree length $l$ (i.e. the sum of all branch lengths), mean branch rate $\bar{r}$, branch rate of all leaf nodes $r$, and relaxed clock standard deviation $\sigma$. We also included the HKY substitution model term $\kappa$. The mixing of $\kappa$ should not be strongly affected by any of the clock model operators, and thus it served as a positive control in each experiment.

Methodologies were benchmarked using one simulated and eight empirical datasets [49–56]. The latter were compiled by Lanfear as "benchmark alignments" (Table 5) [57, 58]. Each methodology was benchmarked for million-states-per-hour using the Intel Xeon Gold 6138 CPU (2.00 GHz). These terms were multiplied by the ESS-per-state across 20 replicates on the New Zealand eScience Infrastructure (NeSI) cluster to compute the total ESS/hr of each dataset under each setting. All methodologies used identical models and operator configurations, except where a difference is specified.

### Round 1: A simple operator-weight learning algorithm greatly improved performance

We compared the nocons, cons, and adapt operator configurations (Table 4). nocons contained all of the standard BEAST 2 operator configurations and weightings for *real*, *cat*, and *quant*. cons additionally contained (cons)tant distance operators and employed the same operator weighting scheme used previously [33] (*real* and *quant* only). Finally, the adapt configuration combined all of the above applicable operators, as well as the simple-but-bold `SampleFromPrior` operator, and learned the weights of each operator using the `AdaptiveOperatorSampler`.

**Table 4. Operator configurations and the substitution rate parameterisations which each operator is applicable to.**

| Configuration | Operator | Weight | *real* | *cat* | *quant* |
|---|---|---|---|---|---|
| nocons | `RandomWalk(`$\vec{\mathcal{R}}$`)` | 10 | ✓ | ✓ | |
| | `Scale(`$\vec{\mathcal{R}}$`)` | 10 | ✓ | | |
| | `Uniform(`$\vec{\mathcal{R}}$`)` | 10 | | ✓ | ✓ |
| | `Interval(`$\vec{\mathcal{R}}$`)` | 10 | | | ✓ |
| | `Swap(`$\vec{\mathcal{R}}$`)` | 10 | ✓ | ✓ | ✓ |
| | `Scale(`$\sigma$`)` | 10 | ✓ | ✓ | ✓ |
| cons | `ConstantDistance(`$\vec{\mathcal{R}}$`,`$t$`)` | $20 \times \frac{2N-2}{2N-1}$ | ✓ | | ✓ |
| | `SimpleDistance(`$\vec{\mathcal{R}}$`,`$t$`)` | $\frac{20}{2} \times \frac{1}{2N-1}$ | ✓ | | ✓ |
| | `SmallPulley(`$\vec{\mathcal{R}}$`)` | $\frac{20}{2} \times \frac{1}{2N-1}$ | ✓ | | ✓ |
| | `RandomWalk(`$\vec{\mathcal{R}}$`)` | 5 | ✓ | | |
| | `Scale(`$\vec{\mathcal{R}}$`)` | 2.5 | ✓ | | |
| | `Uniform(`$\vec{\mathcal{R}}$`)` | 5 | | | ✓ |
| | `Interval(`$\vec{\mathcal{R}}$`)` | 2.5 | | | ✓ |
| | `Swap(`$\vec{\mathcal{R}}$`)` | 2.5 | ✓ | | ✓ |
| | `CisScale(`$\sigma$`,`$\vec{\mathcal{R}}$`)` | 10 | ✓ | | |
| | `Scale(`$\sigma$`)` | 10 | | | ✓ |
| adapt | `AdaptiveOperatorSampler(`$\sigma$`)` | 10 | ✓ | ✓ | ✓ |
| | `AdaptiveOperatorSampler(`$\vec{\mathcal{R}}$`)` | $30 \times \frac{2N-2}{2N-1}$ | ✓ | | ✓ |
| | `AdaptiveOperatorSampler(`$\vec{\mathcal{R}}$`)` | 30 | | ✓ | |
| | `AdaptiveOperatorSampler(root)` | $30 \times \frac{1}{2N-1}$ | ✓ | | ✓ |
| AVMVN | `AdaptiveOperatorSampler(`$\sigma$`)` | 10 | ✓ | | ✓ |
| | `AdaptiveOperatorSampler(leaf)` | $30 \times \frac{N}{2N-1}$ | ✓ | | ✓ |
| | `AdaptiveOperatorSampler(internal)` | $30 \times \frac{N-2}{2N-1}$ | ✓ | | ✓ |
| | `AdaptiveOperatorSampler(root)` | $30 \times \frac{1}{2N-1}$ | ✓ | | ✓ |
| NER{} | `NarrowExchange` | 15 | ✓ | ✓ | ✓ |
| NER | `AdaptiveOperatorSampler(NER)` | 15 | ✓ | | ✓ |

Within each configuration (and substitution rate parameterisation), the weight behind $\vec{\mathcal{R}}$ sums to 30, the weight of $\sigma$ is equal to 10, and the weight of NER is equal to 15. Operators which apply to specific node sets (root, internal, leaf, or all) are weighted according to leaf count $N$. The adaptive operators are further broken down in Table 2. All other operators (i.e. those which apply to which apply to other terms in the state such as the nucleotide substitution model) are held constant within each dataset.

This experiment revealed that nocons usually performed better than cons on smaller datasets (i.e. small *L*) while cons consistently performed better on larger datasets (Fig 8 and S1 Fig). This result is unsurprising (Fig 3). Furthermore, the adapt setup dramatically improved mixing for *real* by finding the right balance between cons and nocons. This yielded an ESS/hr (averaged across all 9 datasets) 95% faster than cons and 520% faster than nocons, with respect to leaf branch rates, and 620% and 190% faster for $\sigma$. Similar results were observed with *quant*. However, adapt neither helped nor harmed *cat*, suggesting that the default operator weighting scheme was sufficient.

This experiment also revealed that the standard `Scale` operator was preferred over `CisScale` for the *real* configuration. Averaged across all datasets, the learned weights behind these two operators were 0.47 and 0.03. This was due to the computationally demanding nature of `CisScale` which invokes the i-CDF function. In contrast, the performance of `Scale` and

## Relaxed clock model optimisation protocol

**Fig 7. Protocol for optimising clock model methodologies.** Each area (detailed in Models and methods) is optimised sequentially, and the best setting from each step is used when optimising the following step.

`CisScale` were more similar under the *quant* configuration and were weighted at 0.28 and 0.45. For both *real* and *quant*, proposals which altered quantiles, while leaving the rates constant (`Scale` and `CisScale` respectively), were preferred.

Overall, the `AdaptiveOperatorSampler` operator was included in all subsequent rounds in the tournament.

**Table 5. Benchmark datasets, sorted in increasing order of taxon count *N*.**

|  | *N* | *P* | *L* (kb) | $L_{unq}$ (kb) | $\hat{\sigma}$ | Description |
|---|---|---|---|---|---|---|
| 1 | 38 | 16 | 15.5 | 10.5 | 0.44 | Seed plants (Ran 2018 [49]) |
| 2 | 44 | 7 | 5.9 | 1.8 | 0.30 | Squirrel Fishes (Dornburg 2012 [50]) |
| 3 | 44 | 3 | 1.9 | 0.8 | 0.27 | Bark beetles (Cognato 2001 [51]) |
| 4 | 51 | 6 | 5.4 | 1.8 | 0.52 | Southern beeches (Sauquet 2011 [52]) |
| 5 | 61 | 8 | 6.9 | 4.3 | 0.53 | Bony fishes (Broughton 2013 [53]) |
| 6 | 70 | 3 | 2.2 | 0.9 | 0.19 | Caterpillars (Kawahara 2013 [54]) |
| 7 | 80 | 1 | 10.0 | 0.9 | 0.82 | Simulated data |
| 8 | 94 | 4 | 2.2 | 1 | 0.34 | Bees (Rightmyer 2013 [55]) |
| 9 | 106 | 1 | 0.8 | 0.5 | 0.37 | Songbirds (Moyle 2016 [56]) |

Number of partitions *P*, total alignment length *L*, and number of unique site patterns $L_{unq}$ in the alignment are specified. Clock standard deviation estimates $\hat{\sigma}$ are of moderate magnitude, suggesting that most of these datasets are not clock-like.

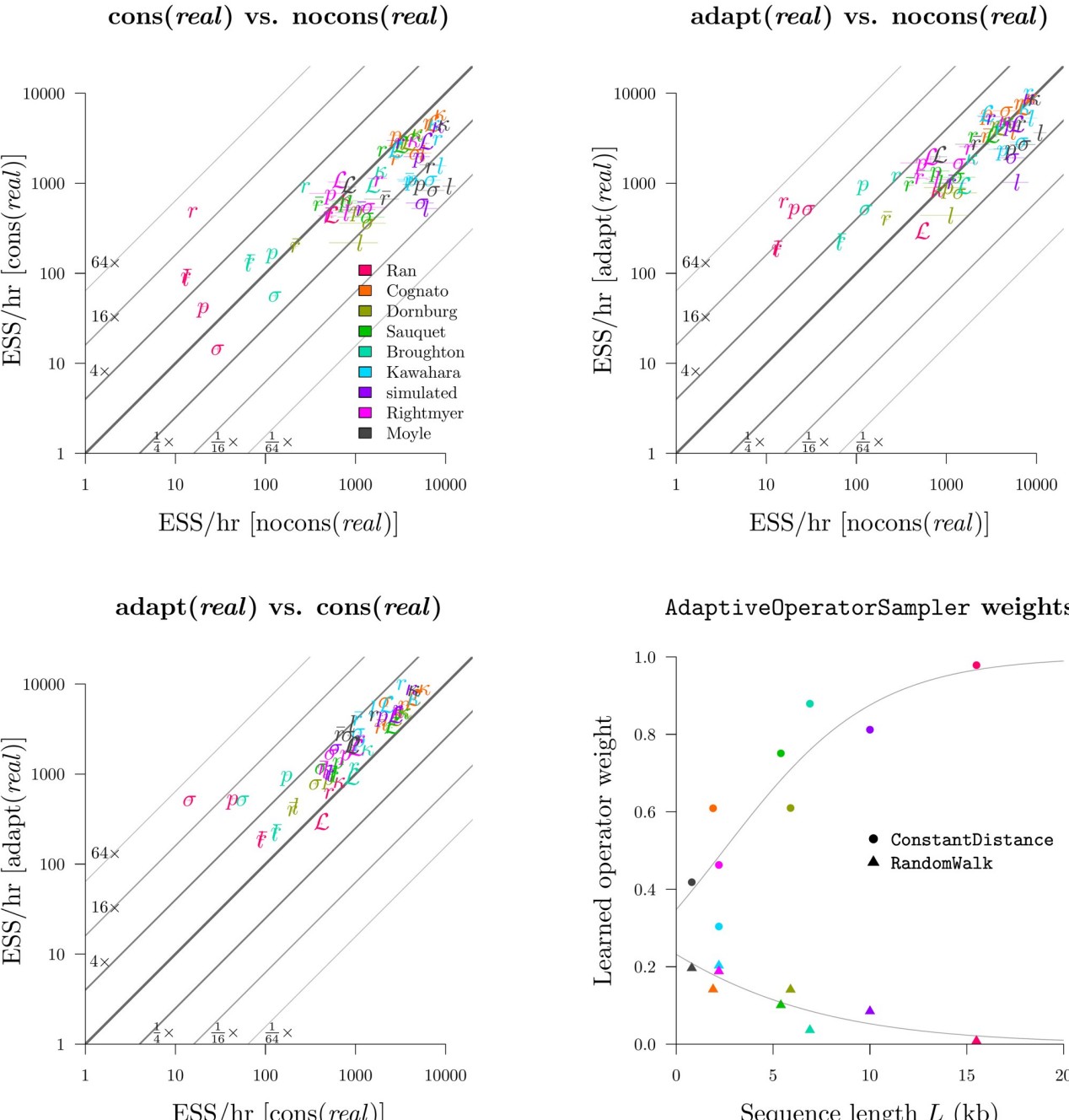

**Fig 8. Round 1: Benchmarking the `AdaptiveOperatorSampler` operator.** Top left, top right, bottom left: each plot compares the ESS/hr (±1 standard error) across two operator configurations. Bottom right: the effect of sequence length $L$ on operator weights learned by `AdaptiveOperatorSampler`. Both sets of observations are fit by logistic regression models. The benchmark datasets are displayed in Table 5. The *cat* and *quant* settings are evaluated in S1 Fig.

## Round 2: The *real* parameterisation yielded the fastest mixing

In Round 1 we selected the best configuration for each of the three rate parameterisations described in Branch rate parameterisations, and in Round 2 we compared the three to each other. adapt (*real*) and adapt (*quant*) both employed constant distance tree operators [33] and

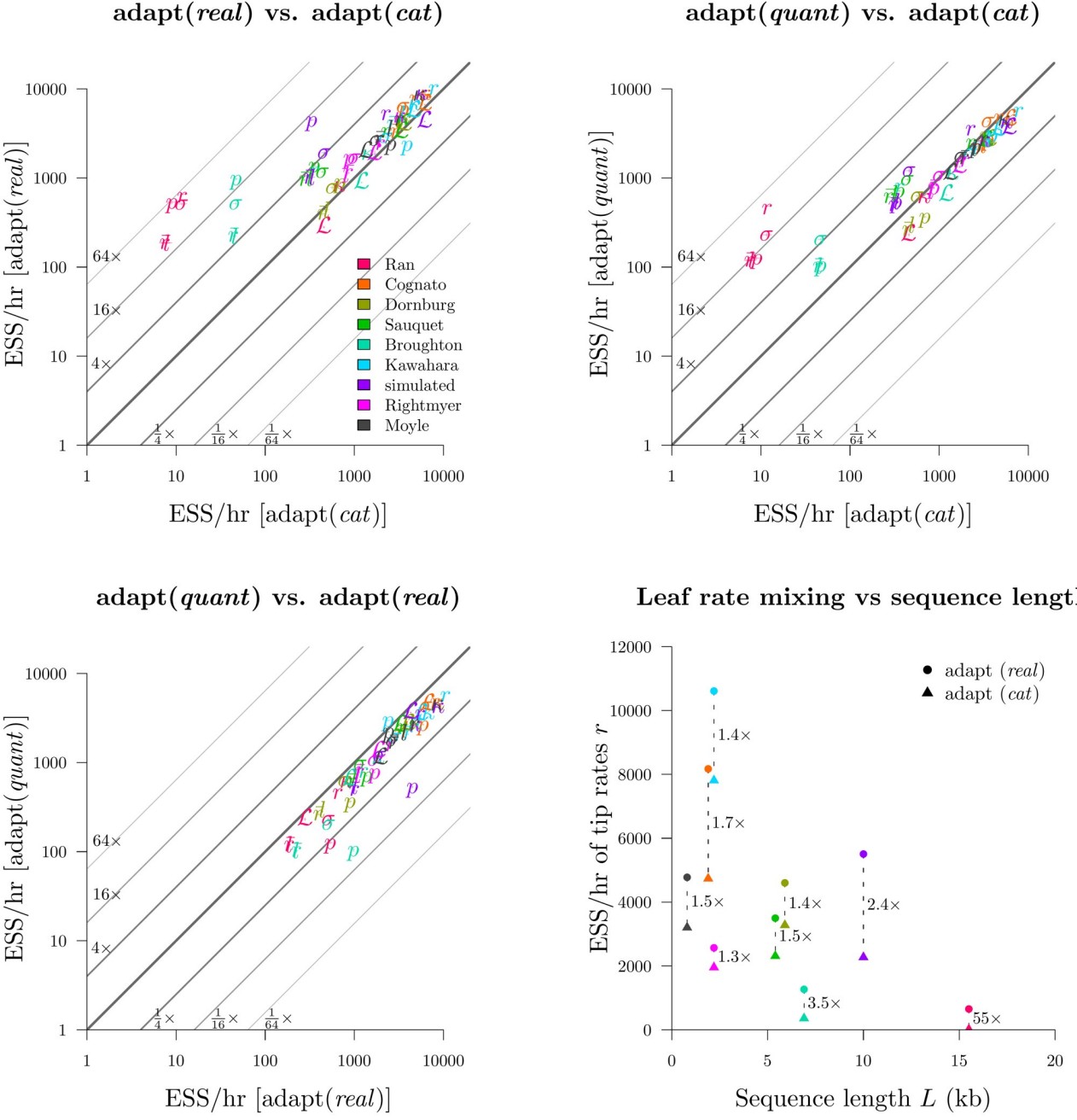

**Fig 9. Round 2: Benchmarking substitution rate parameterisations.** Top left, top right, bottom left: the adapt (*real*), adapt (*cat*), and adapt (*quant*) configurations were compared. Bottom right: comparison of the mean tip substitution rate ESS/hr as a function of alignment length *L*.

both used the `AdaptiveOperatorSampler` operator to learn clock model operator weights. Clock model operators weights were also learned in the adapt (*cat*) configuration.

This experiment showed that the *real* parameterisation greatly outperformed *cat* on most datasets and most parameters (Fig 9). This disparity was strongest for long alignments. In the most extreme case, leaf substitution rates *r* and clock standard deviation $\sigma$ both mixed around 50× faster on the 15.5 kb seed plant dataset (Ran et al. 2018 [49]) for *real* than they did for *cat*.

**Fig 10. Comparison of runtimes across methodologies.** The computational time required for a setting to sample a single state is divided by that of the nocons (*cat*) configuration. The geometric mean under each configuration, averaged across all 9 datasets, is displayed as a horizontal bar.

The advantages in using constant distance operators would likely be even stronger for larger *L*. Furthermore, *real* outperformed *quant* on most datasets, but this was mostly due to the slow computational performance of *quant* compared with real, as opposed to differences in mixing prowess (Fig 10). Irrespective of mixing ability, the adapt (*real*) configuration had the best computational performance and generated samples 40% faster than adapt (*cat*) and 60% faster than adapt (*quant*).

Overall, we determined that *real*, and its associated operators, made the best parameterisation covered here and it proceeded to the following rounds of benchmarking. This outcome allowed Rounds 3, 4, and 5 to commence. If *cat* dominated in this round, then the operators benchmarked in Rounds 4 and 5 would not be applicable to the parameterisation and the Bactrian proposal kernel benchmarked in Round 3 would not be applicable to branch rate categories in its present form.

## Round 3: Bactrian proposal kernels were around 15% more efficient than uniform kernels

We benchmarked the adapt (*real*) configuration with a) standard uniform proposal kernels, and b) Bactrian(0.95) kernels [31]. These kernels applied to all clock model operators. These results confirmed that the Bactrian kernel yields faster mixing than the standard uniform kernel (Fig 11). All relevant continuous parameters considered had an ESS/hr, averaged across the 9 datasets, between 15% and 20% faster compared with the standard uniform kernel. Although the Bactrian proposal made little-to-no difference to the caterpillar dataset

## Comparison of Bactrian+adapt(*real*) and adapt(*real*)

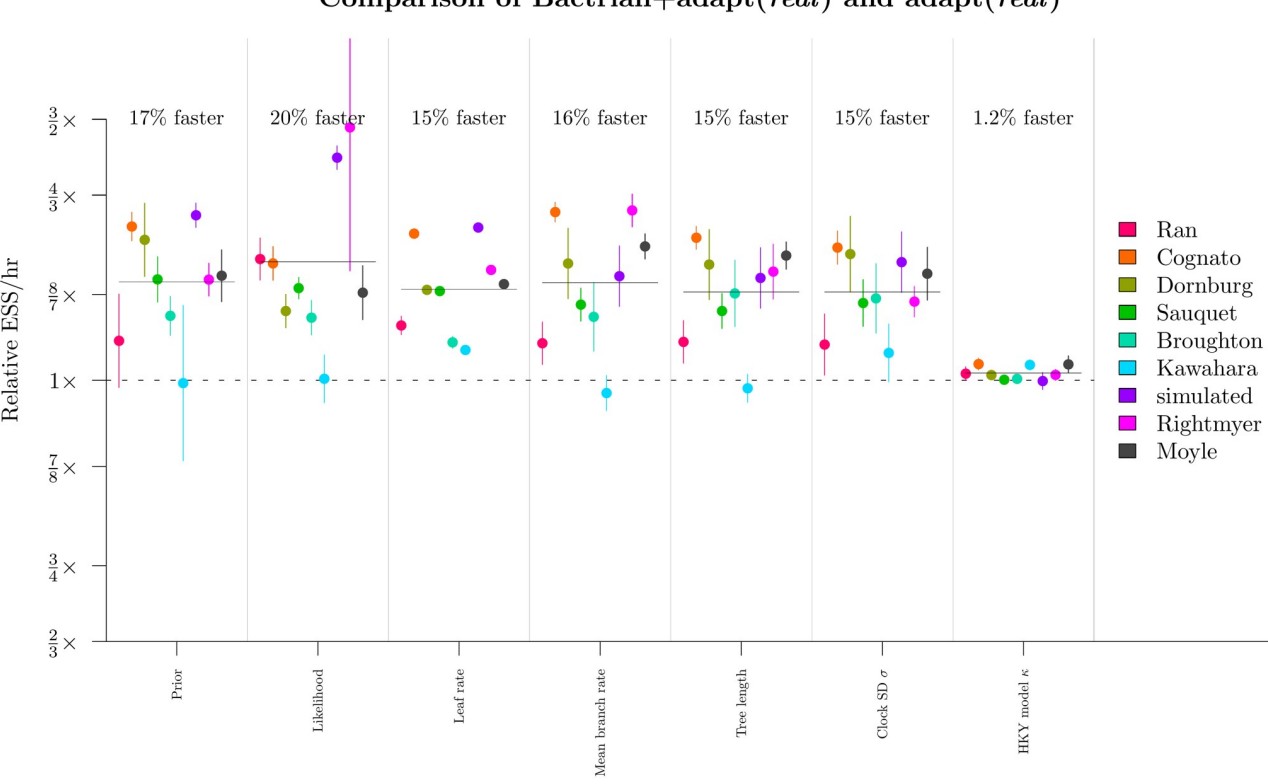

**Fig 11. Round 3: Benchmarking the Bactrian kernel.** The ESS/hr (±1 s.e.) under the Bactrian configuration, divided by that under the uniform kernel, is shown in the y-axis for each dataset and relevant parameter. Horizontal bars show the geometric mean under each parameter.

(Kawahara and Rubinoff 2013 [54]), every other dataset did in fact benefit. Bactrian proposal kernels proceeded to round 4 of the relaxed clock model optimisation protocol.

### Round 4: NER operators outperformed on larger datasets

Our initial screening of the `NarrowExchangeRate` (NER) operators revealed that the NER$\{\mathcal{D}_{AE}, \mathcal{D}_{BE}, \mathcal{D}_{CE}\}$ operator outperformed the standard `NarrowExchange` / NER{} operator about 25% of the time on simulated data, however it was also very sensitive to the dataset. Therefore we wrapped up the two operators (NER{} and NER$\{\mathcal{D}_{AE}, \mathcal{D}_{BE}, \mathcal{D}_{CE}\}$) within an `AdaptiveOperatorSampler` operator so that the appropriate weights could be learned. In this round we benchmarked the Bactrian + adapt (*real*) setting with the adaptive NER operator (Table 4). The benchmark datasets are fairly non clock-like and therefore could potentially benefit from NER (Table 5).

Our experiments confirmed that NER$\{\mathcal{D}_{AE}, \mathcal{D}_{BE}, \mathcal{D}_{CE}\}$ was indeed superior on larger datasets (where $L > 5$kb; Fig 12). While there was no significant difference in the ESS/hr of continuous parameters, NER$\{\mathcal{D}_{AE}, \mathcal{D}_{BE}, \mathcal{D}_{CE}\}$ did have an acceptance rate 41% higher than that of the standard `NarrowExchange` operator in the most extreme case (the bony fish alignment by Broughton et al. [53]). The moderate variance in branch substitution rates ($\hat{\sigma} = 0.5$), coupled with a long alignment (7kb), and high topological uncertainty (Fig 12) made this dataset the perfect target. Every acceptance of a branch rearrangement proposal yields a new topology and thus facilitates traversal of tree space.

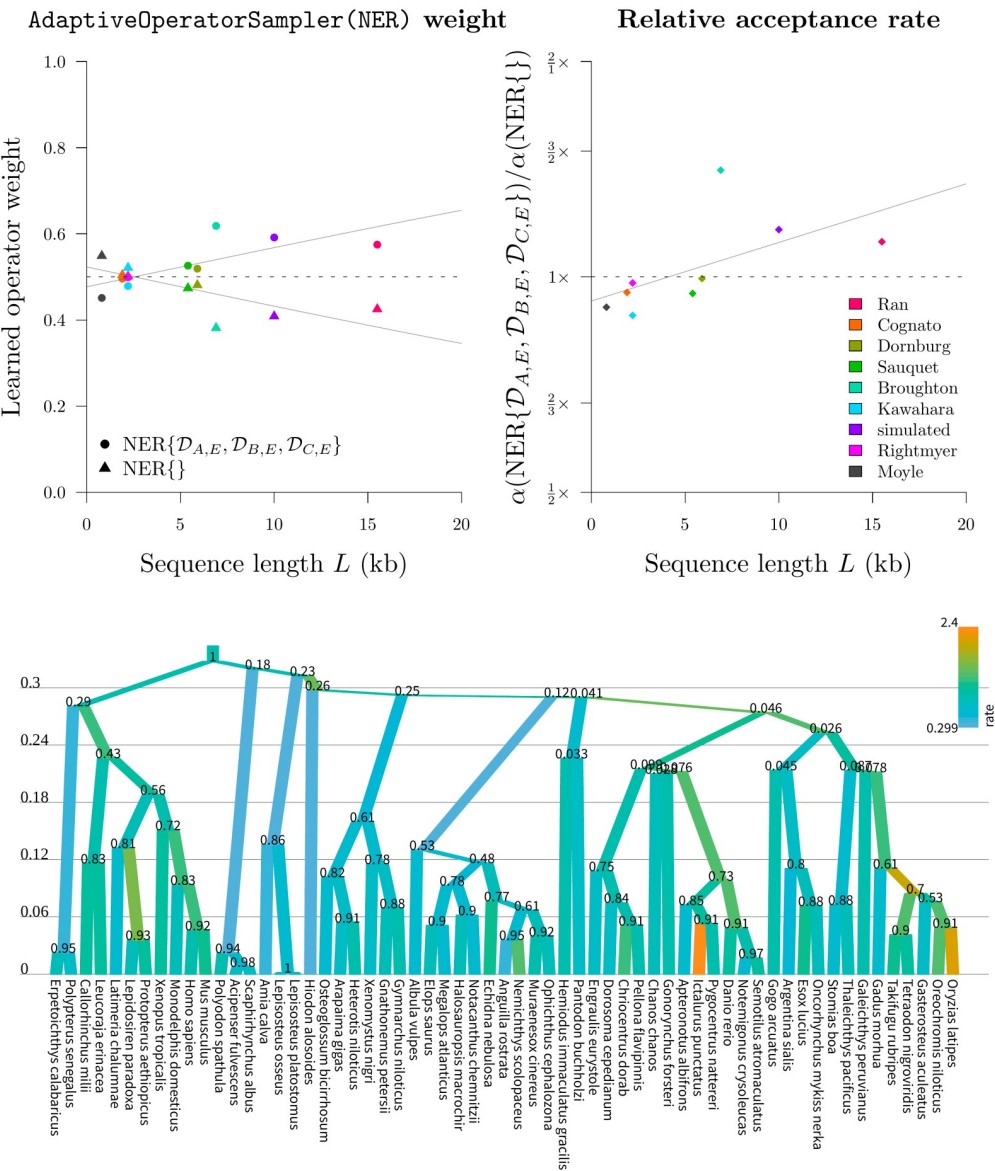

**Fig 12. Round 4: Benchmarking the NER operators.** Top: the learned weights (left) behind the two NER operators (NER{} and NER{$\mathcal{D}_{A,E}, \mathcal{D}_{B,E}, \mathcal{D}_{C,E}$}), and the relative difference between their acceptance rates $\alpha$ (right), are presented as functions of sequence length. Logistic and logarithmic regression models are shown, respectively. Bottom: maximum clade credibility tree of the bony fish dataset by Broughton et al. 2013 [53]. This alignment received the strongest boost from NER, likely due to its high topological uncertainty and branch rate variance. Branches are coloured by substitution rate, the y-axis shows time units, and internal nodes are labelled with posterior clade support. Tree visualised using UglyTrees [59].

In contrast, the standard `NarrowExchange` operator outperformed on smaller datasets. The new operator was not always helpful and sometimes it even hindered performance. Use of an adaptive operator (`AdaptiveOperatorSampler`) removes the burden from the user in making the decision of which operator to use. The `AdaptiveOperatorSampler` `(NER)` operator proceeded into the final round of the tournament.

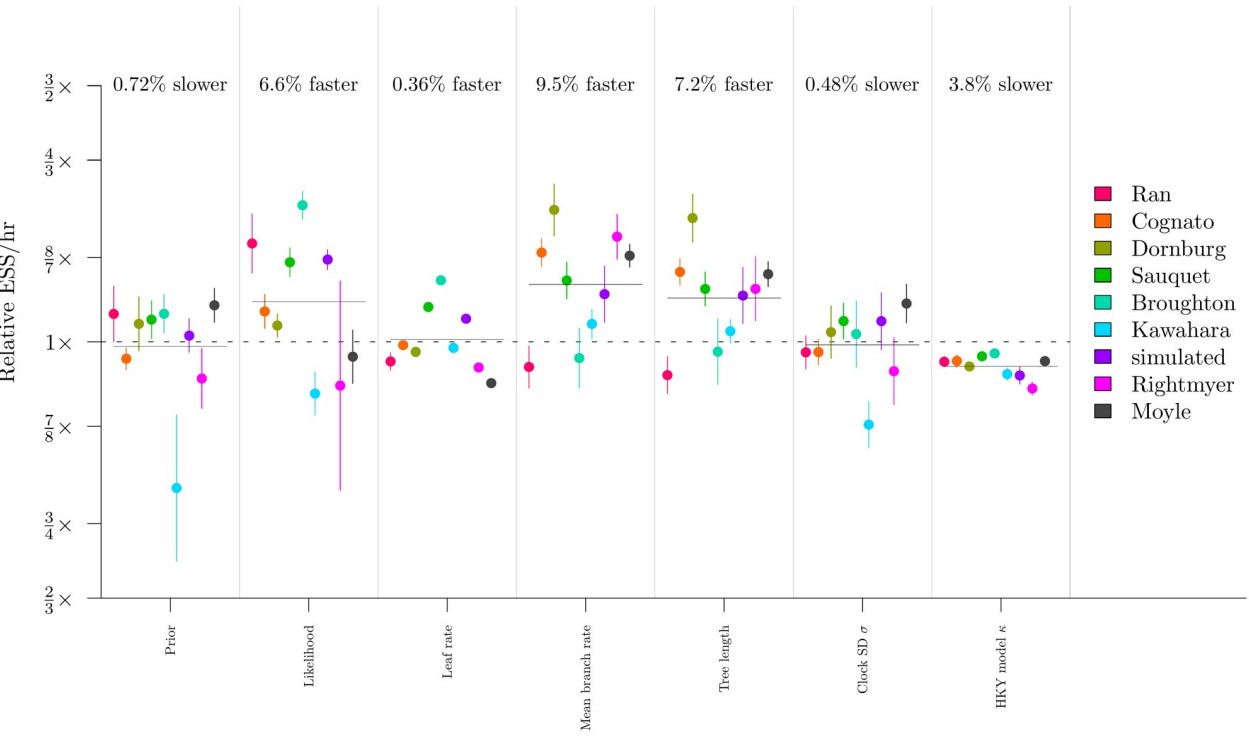

**Fig 13. Round 5: Benchmarking the `LeafAVMVN` operator.** See Fig 11 caption for figure notation.

### Round 5: The AVMVN leaf rate operator was computationally demanding and improved mixing very slightly

We tested the applicability of the AVMVN kernel to leaf rate proposals. This operator exploits any correlations which exist between leaf branch substitution rates. To do this, we wrapped the `LeafAVMVN` operator within an `AdaptiveOperatorSampler` (Table 2). The two configurations compared here were a) adapt + Bactrian + NER (*real*) and b) AVMVN + NER + Bactrian + adapt (*real*) (Table 4).

These results showed that the AVMVN operator yielded slightly better mixing (around 6% faster) for the tree likelihood, the tree length, and the mean branch rate (Fig 13). However, it also produced slightly slower mixing for $\kappa$, reflecting the high computational costs associated with the `LeafAVMVN` operator (Fig 10). The learned weight of the `LeafAVMVN` operator was quite small (ranging from 1 to 8% across all datasets), again reflecting its costly nature, but also reinforcing the value in having an adaptive weight operator which penalises slow operators. The `LeafAVMVN` operator provided some, but not much, benefit in its current form.

Overall, we determined that the AVMVN operator configuration was the final winner of the tournament, however its performance benefits were minor and therefore the computational complexities introduced by the `LeafAVMVN` operator may not be worth the trouble.

### Tournament conclusion

In conjunction with all settings which came before it, the tournament winner outperformed both the historical *cat* configuration [3] as well as the recently developed cons (*real*) scheme [33]. Averaged across all datasets, this configuration yielded a relaxed clock mixing rate

between 1.2 and 13 times as fast as *cat* and between 1.8 and 7.8 times as fast as cons (*real*), depending on the parameter. For the largest dataset considered (seed plants by Ran et al. 2018 [49]), the new settings were up to 66 and 37 times as fast respectively. This is likely to be even more extreme for larger alignments.

## Discussion

### Modern operator design

Adaptability and advanced proposal kernels, such as Bactrian kernels, are increasingly prevalent in MCMC operator design [60–63]. Adaptive operators undergo training to improve their efficiency over time [64]. In previous work, the conditional clade probabilities of neighbouring trees have served as the basis of adaptive tree operators [26, 27]. Proposal step sizes can be tuned during MCMC [38]. The mirror kernel learns a target distribution which acts as a "mirror image" of the current point [32]. The AVMVN operator learns correlations between numerical parameters in order to traverse the joint posterior distribution efficiently [30].

Here, we introduced an adaptive operator which learns the weights of other operators, by using a target function that rewards operators which bring about large changes to the state and penalises operators which exhibit poor computational runtime (Eq 11). We demonstrated how learning the operator weightings, on a dataset-by-dataset basis, can improve mixing by up to an order of magnitude. We also demonstrated the versatility of this operator by applying it to a variety of settings. Assigning operator weights is an important task in Bayesian MCMC inference and the use of such an operator can relieve some of the burden from the person making this decision. However, this operator is no silver bullet and it must be used in a way that maintains ergodicity within the search space [64].

We also found that a Bactrian proposal kernel quite reliably increased mixing efficiency by 15–20% (Fig 11). Similar observations were made by Yang et al. [31]. While this may only be a modest improvement, incorporation of the Bactrian kernels into pre-existing operators is a computationally straightforward task and we recommend implementing them in Bayesian MCMC software packages.

### Traversing tree space

In this article we introduced the family of narrow exchange rate operators (Fig 5). These operators are built on top of the narrow exchange operator and are specifically designed for the (uncorrelated) relaxed clock model, by accounting for the correlation which exists between branch lengths and branch substitution rates. This family consists of 48 variants, each of which conserves a unique subset of genetic distances before and after the proposal. While most of these operators turned out to be worse than narrow exchange, a small subset were more efficient, but only on large datasets.

Lakner et al. 2008 categorised tree operators into two classes. "Branch-rearrangement" operators relocate a branch and thus alter tree topology. Members of this class include narrow exchange, nearest neighbour interchange, and subtree-prune-and-regraft [43]. Whereas "branch-length" operators propose branch lengths, but can potentially alter the tree topology as a side-effect. Such operators include subtree slide [65], LOCAL [66], and continuous change [67]. Lakner et al. 2008 observed that topological proposals made by the former class consistently outperformed topological changes invoked by the latter [68].

We hypothesised that the increased efficiency behind narrow exchange rate operators could facilitate proposing internal node heights in conjunction with branch rearrangements. This would enable the efficient exploration of both topology and branch length spaces with a single proposal. Unfortunately, by incorporating a random walk on the height of the node being

relocated, the acceptance rate of the operator declined dramatically (Fig 6). This decline was greater when more genetic distances were conserved.

These findings support Lakner's hypothesis. The design of operators which are able to efficiently traverse topological and branch length spaces simultaneously remains an open problem.

## Larger datasets require smarter operators

As signal within the dataset becomes stronger, the posterior distribution becomes increasingly peaked (Fig 3). This change in the posterior topology necessitates the use of operators which exploit known correlations in the posterior density; operators such as AVMVN [30], constant distance [33], and the narrow exchange rate operators introduced in this article.

We have shown that while the latter two operators are efficient on large alignments, they are also quite frequently outperformed by simple random walk operators on small alignments. For instance, we found that constant distance operators outperformed standard operator configurations by up to two orders of magnitude on larger datasets but they were up to three times slower on smaller ones (Fig 8). Similarly, our narrow exchange rate operators were up to 40% more efficient on large datasets but up to 10% less efficient on smaller ones (Fig 12).

This emphasises the value in our adaptive operator weighting scheme, which can ensure that operator weights are suitable for the size of the alignment. Given the overwhelming availability of sequence data, high performance on large datasets is more important than ever.

## Future outlook

Based on these experiments, we have corroborated the results of Zhang and Drummond 2020 [33] by showing that the relaxed clock rate parameterisation can be greatly superior to the categories parameterisation. However this is conditional on the use of operators. Standard scale operators are ineffective in the real rate space on large datasets and constant distance operators tend to be the same on smaller ones. As the categories configuration is the current relaxed clock model default in both BEAST as well as BEAST 2, we therefore recommend researchers make the transition to the real rate parameterisation and employ an operator setup similar to that described in this article.

However, there remain several avenues for improvement. First, the use of a Weibull clock model prior, as opposed to a log-normal, could further decrease calculation time due to its closed-form inverse cumulative distribution function. Second, the leaf rate AVMVN operator presented in this article could be improved by proposing rates for only a small subset of correlated branch rates per proposal, as opposed to all at once. This would mitigate the computational burden associated with a covariance matrix which grows with the tree size (An adaptive leaf rate operator). Finally, while our adaptive weighting operator can learn operator weights within parameter spaces, it is not clear how to best assign weights between spaces; the relative weighting between clock model operators and tree topology operators for instance.

## Conclusion

In this article, we delved into the highly correlated structure between substitution rates and divergence times of relaxed clock models, in order to develop MCMC operators which traverse its posterior space efficiently. We introduced a range of relaxed clock model operators and compared three molecular substitution rate parameterisations. These methodologies were compared by constructing phylogenetic models from several empirical datasets and comparing their abilities to converge in a tournament-like protocol (Fig 7). The methods introduced are adaptive, treat each dataset differently, and rarely perform worse than without adaptation.

This work has produced an operator configuration which is highly effective on large alignments and can explore relaxed clock model space up to two orders of magnitude more efficiently than previous setups.

## Supporting information

**S1 Appendix. Rate quantiles and operators.** The linear piecewise approximation used in the *quant* parameterisation is described. Constant distance tree operators, `CisScale`, and `NarrowExchangeRate` are extended to the *quant* parameterisation. A second NER algorithm is specified.
(PDF)

**S2 Appendix. Well-calibrated simulation studies.** Methodologies are validated using well-calibrated simulation studies.
(PDF)

**S1 Fig. Round 1: Benchmarking adapt under the *quant* and *cat* parameterisations.** These results show that *cat* does not benefit from adaptive weight sampling. Whereas, adapt and cons both greatly improve the *quant* parameterisation for most datasets, as expected.
(PDF)

## Acknowledgments

We wish to thank Alexei J. Drummond for his help during conception of the narrow exchange rate operators.

## Author Contributions

**Conceptualization:** Jordan Douglas, Remco Bouckaert.

**Data curation:** Jordan Douglas.

**Formal analysis:** Jordan Douglas, Remco Bouckaert.

**Funding acquisition:** Remco Bouckaert.

**Investigation:** Jordan Douglas, Remco Bouckaert.

**Methodology:** Jordan Douglas, Rong Zhang, Remco Bouckaert.

**Software:** Jordan Douglas, Rong Zhang, Remco Bouckaert.

**Supervision:** Remco Bouckaert.

**Validation:** Jordan Douglas, Rong Zhang.

**Visualization:** Jordan Douglas.

**Writing – original draft:** Jordan Douglas.

**Writing – review & editing:** Jordan Douglas, Rong Zhang, Remco Bouckaert.

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
