## [Decision Letter · Decision Letter 0]

29 Oct 2020

Dear Mr Douglas,

Thank you very much for submitting your manuscript "Adaptive dating and fast proposals: revisiting the phylogenetic relaxed clock model" for consideration at PLOS Computational Biology. As with all papers reviewed by the journal, your manuscript was reviewed by members of the editorial board and by several independent reviewers. The reviewers appreciated the attention to an important topic. Based on the reviews, we are likely to accept this manuscript for publication, providing that you modify the manuscript according to the review recommendations.

Sincerely,

Roger Dimitri Kouyos

Associate Editor

PLOS Computational Biology

Stefano Allesina

Deputy Editor

PLOS Computational Biology

[LINK]

Reviewer's Responses to Questions

**Comments to the Authors:**

Reviewer #1: This paper describes a number of measures to improve the efficiency and performance of Bayesian phylogenetic analysis using relaxed molecular clocks. These measures are evaluated using a sequential approach, and benchmarked using synthetic data and eight empirical datasets. The authors find considerable improvements in MCMC efficiency, particularly for large datasets. One especially useful finding is that an adaptive operator weighting scheme can tune the operator weights to optimise performance for datasets of different sizes.

I have no major concerns about the parameterisations and operators that are proposed in the paper, but the paper itself needs a reduction in length. There is an excessive number of figures and tables, and the level of detail in the Methods and Results can be cut down.

In the Abstract, I suggest adding a brief explanation of uncorrelated relaxed clocks (for readers unfamiliar with the term).

In the Author Summary, I suggest mentioning molecular dating or divergence time estimation, which is the main aim when using relaxed clock models.

In the first paragraph of the Introduction, it should be noted that the molecular clock is only essential to molecular dating, but not to phylogenetic analysis in general. Clock models are rarely used in phylogenetic analysis unless one of the goals is to infer the divergence times. BEAST is a notable exception because it always implements some form of clock model. In contrast, the clock models in MrBayes are quite rarely used.

In the fourth paragraph of the Introduction, it would be reasonable to cite the work of Thorne et al. (1998, Mol Biol Evol), who presented the first Bayesian relaxed (autocorrelated) clock and also described a likelihood approximation to reduce computational burden. One other consideration is whether autocorrelated or uncorrelated models are more biologically relevant. This has been considered in a number of studies, including by Lepage et al. (2007, cited by the authors).

In the first paragraph of the Methods, presumably theta represents the parameters of the branching or coalescent model used to generate the tree prior. It would be useful to clarify this.

In the first paragraph of the section Clock Model Operators, please include an explanation of how the parameter s learned over the course of the MCMC. Is this tuned to achieve a target acceptance rate?

Combine some of the Tables listing operators?

Given the superior performance of the _real_ parameterisation for the rate, should this be adopted as the default in BEAST rather than _cat_?

Figure 13 is only mentioned briefly in the text and does not seem essential.

In the Discussion it would be helpful for the authors to describe some potential avenues for improving the proposed operator configuration.

Reviewer #2: Review is uploaded as an attachment

**Have all data underlying the figures and results presented in the manuscript been provided?**

Reviewer #1: **No: **Numerical data from figures unavailable

Reviewer #2: Yes

PLOS authors have the option to publish the peer review history of their article (what does this mean?). If published, this will include your full peer review and any attached files.

Reviewer #1: No

Reviewer #2: No
---

## [Decision Letter · Decision Letter 1]

30 Nov 2020

Dear Mr Douglas,

We are pleased to inform you that your manuscript 'Adaptive dating and fast proposals: revisiting the phylogenetic relaxed clock model' has been provisionally accepted for publication in PLOS Computational Biology.

Best regards,

Roger Dimitri Kouyos

Associate Editor

PLOS Computational Biology

Stefano Allesina

Deputy Editor

PLOS Computational Biology

Please address the few remaining issues regarding labelling and figure numbering (highlighted by reviewer 2) in the final version of the paper.

Reviewer's Responses to Questions

**Comments to the Authors:**

Reviewer #1: The authors have addressed all of my previous comments. The new "Future Outlook" section at the end of the Discussion is particularly useful.

I have no further concerns about the study or further comments on the manuscript.

Reviewer #2: Dear authors, Thank you for the revision and responses. The authors have addressed well the reviewers' questions and comments.

Before the final submission, please note there are still several inconsistencies in the node height abbreviation (Table 2, 3, and 4) and a table number is missing (page 12).

**Have all data underlying the figures and results presented in the manuscript been provided?**

Reviewer #1: Yes

Reviewer #2: Yes

PLOS authors have the option to publish the peer review history of their article (what does this mean?). If published, this will include your full peer review and any attached files.

Reviewer #1: No

Reviewer #2: No

---

## [Editor Report · Acceptance letter]

27 Jan 2021

PCOMPBIOL-D-20-01606R1 

Adaptive dating and fast proposals: revisiting the phylogenetic relaxed clock model

Dear Dr Douglas,

I am pleased to inform you that your manuscript has been formally accepted for publication in PLOS Computational Biology. Your manuscript is now with our production department and you will be notified of the publication date in due course.

With kind regards,

Alice Ellingham
